# Variational inference via Gaussian interacting particles in the Bures–Wasserstein geometry

**Giacomo Borghi** [1]  **José A. Carrillo** [2]

## Abstract

Motivated by variational inference methods, we propose a zeroth-order algorithm for solving optimization problems in the space of Gaussian probability measures. The algorithm is based on an interacting system of Gaussian particles that stochastically explore the search space and self-organize around global minima via a consensus-based optimization (CBO) mechanism. Its construction relies on the Linearized Bures–Wasserstein (LBW) space, a novel parametrization of Gaussian measures we introduce for efficient computations. We establish well-posedness and study the convergence properties of the particle dynamics via a mean-field approximation. Numerical experiments on variational inference tasks demonstrate the algorithm's robustness and superior performance with respect to deterministic gradient-based method in presence of low-dimensional non log-concave targets.

## 1. Introduction

**Problem.** Given a target probability measure $\mu^{\text{targ}} \in \mathcal{P}(\mathbb{R}^d)$ a widespread problem in statistics and computational sciences consists of finding

$$\mu^\star \in \operatorname*{argmin}_{\mu \in \Pi} \mathcal{E}(\mu) \qquad (1)$$

where $\Pi$ is a family of parametrized probability measures, and $\mathcal{E}(\cdot) = \mathcal{D}(\cdot \mid \mu^{\text{targ}})$ is a functional which quantifies the discrepancy from a target. A classical example is the Bayesian inference problem where $\mu^{\text{targ}}(\mathrm{d}x) \propto$

---

[1]Maxwell Institute for Mathematical Sciences and Department of Mathematics, School of Mathematical and Computer Sciences, Heriot-Watt University, Edinburgh, UK [2]Mathematical Institute, University of Oxford, Oxford, UK. Correspondence to: Giacomo Borghi <g.borghi@hw.ac.uk>, José A. Carrillo <jose.carrillo@maths.ox.ac.uk>.

*Proceedings of the 43$^{rd}$ International Conference on Machine Learning*, Seoul, South Korea. PMLR 306, 2026. Copyright 2026 by the author(s).

$\exp(-V(x))\mathrm{d}x$ for some $V : \mathbb{R}^d \to \mathbb{R}$, and the discrepancy measure corresponds to the Kullback–Leibler divergence, $\mathcal{E}(\cdot) = \mathrm{KL}(\cdot \mid \mu^{\text{targ}})$ (Blei et al., 2017). The settings we considered though, are the more general one of Variational Inference (VI), where $\mathcal{E}$ can be given, for instance, by the Maximum Mean Discrepancy, $f$-divergencies, $\chi^2$-divergence, Rényi's $\alpha$-divergence (Blei et al., 2017; Knoblauch et al., 2022).

In this work we consider the Gaussian VI problems (Diao et al., 2023; Katsevich & Rigollet, 2024) where $\Pi \subset \mathcal{P}(\mathbb{R}^d)$ is the set of $d$-dimensional normal distributions

$$\Pi = \mathcal{N}^d := \left\{ \mu = \mathcal{N}(m, \Sigma) \mid m \in \mathbb{R}^d, \Sigma \in \mathrm{Sym}_d^+ \right\}$$

with $\mathrm{Sym}_d^+$ being the space of symmetric positive semi-definite matrices. This approach offers a computationally efficient alternative to traditional Bayesian inference by approximating posterior distributions with tractable Gaussian families, enabling faster approximate inference compared to Markov Chain Monte Carlo methods (Spokoiny & Panov, 2025; Katsevich & Rigollet, 2024).

**State of the art.** Different works, see, for instance, (Alvarez-Melis et al., 2022; Lambert et al., 2022; Diao et al., 2023), propose first-order algorithms as suitable discretization of gradient flows in the space probability measures (Ambrosio et al., 2008) equipped with the $L^2$-Wasserstein distance. When restricting to the space of non-degenerate Gaussians, the distance is known as the Bures–Wasserstein (BW) distance and can be derived from a suitable Riemannian structure (Bhatia et al., 2019; Malagò et al., 2018). First-order algorithms with respect to the Hellinger–Kantorovich metric have been proposed in (Liero et al., 2025b).

Such optimization dynamics can be conveniently written as a system of ODEs for the mean $m \in \mathbb{R}^d$ and the covariance matrix $\Sigma \in \mathrm{Sym}_d^+$ and have shown superior performance with respect to other Gaussian Bayesian inference methods such as the Laplace method (Tierney & Kadane, 1986), see (Lambert et al., 2022). As in Euclidean optimization, though, convexity of objective functional plays a central role in the convergence of gradient-based methods, and the optimizing dynamics may get stuck in local minima, if present.

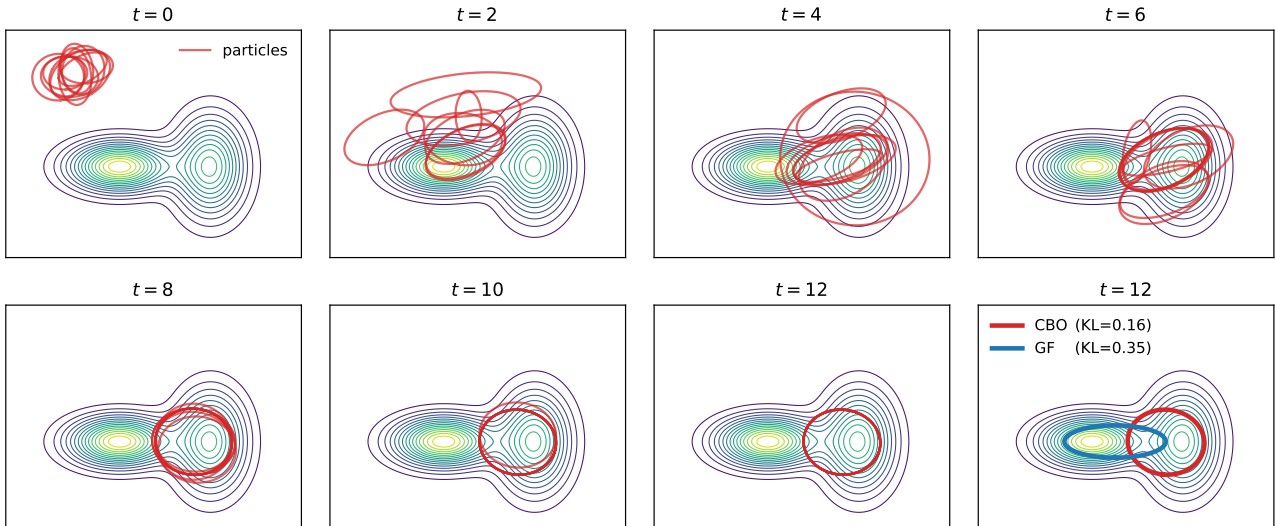

*Figure 1.* Evolution of $N = 10$ Gaussian particles for minimization of KL divergence from a bi-modal target measure (shown by contour lines). Particles evolve according to the CBO-type dynamics proposed in this paper for problems of the form (1) (see Section 3). The final snapshot compares the solution computed by the proposed CBO method with the BW Gradient Flow solution (Lambert et al., 2022); the corresponding KL values are also reported.

**Our contributions using interacting particles.** We propose a novel global, gradient-free optimization algorithm for (1) based on a system of Gaussian particles. Particles follows a Consensus-Based Optimization (CBO) dynamics which exploits the Bures–Wasserstein geometry of $\mathcal{N}^d$.

**Starting point.** In the Euclidean CBO algorithm (Pinnau et al., 2017), each particle stochastically moves towards a consensus point consisting of a weighted average of the entire ensemble, that is considered a proxy for the minimizer. In (Borghi et al., 2025) an algorithm where each particle is a probability measure was proposed by substituting averages with Wasserstein barycenters (Agueh & Carlier, 2011).

In Gaussian settings, every particle is a Gaussian measure

$$\mu_t^i = \mathcal{N}(m_t^i, \Sigma_t^i), \qquad \text{for } i \in [N], \; t \geq 0$$

and, since the $L^2$-Wasserstein barycenter is still a Gaussian, the consensus dynamics of (Borghi et al., 2025) reads

$$\begin{cases} (\overline{m}_t, \overline{\Sigma}_t) := \text{Barycenter}^\omega \left\{ (m_t^i, \Sigma_t^i) \mid i \in [N] \right\} \\ \dot{m}_t^i = \overline{m}_t - m_t^i \qquad i \in [N] \\ \dot{\Sigma}_t^i = \overline{\Sigma}_t(\Sigma_t^i \overline{\Sigma}_t)^{-\frac{1}{2}} - I \qquad i \in [N]. \end{cases} \quad (2)$$

Above, the barycenter serves as the consensus point and is computed according to the Gibbs weight function

$$\omega(\mu) := \exp\left(-\alpha \mathcal{E}(\mu)\right) \qquad \alpha \gg 1, \quad (3)$$

and the tangent vector $\overline{\Sigma}_t(\Sigma_t^i \overline{\Sigma}_t)^{-\frac{1}{2}} - I$ corresponds the optimal transport maps from $\Sigma_t^i$ to $\overline{\Sigma}_t$ (see Appendix A).

Consensus dynamics in Wasserstein spaces have been also formulated in (Bishop & Doucet, 2014; 2021; Cisneros-Velarde & Bullo, 2023) with aim of deriving a distributed algorithm for the computation of Wasserstein barycenters, and not in the context of optimization.

**Contributions.** While coherent with the BW geometry, the particle system (2) lacks stochasticity and is computationally expensive, making it unsuitable for solving the optimization problem (1). To address this,

i) We propose the Linearized Bures–Wasserstein space (LBW), a novel parametrization of the space of Gaussian measures that enables efficient computation and the use of stochastic analysis, while retaining key features of the BW geometry.

ii) We design an efficient CBO-type dynamics in the LBW space to solve (1) (see Figure 1 for an intuition). The particle method is gradient-free and only requires evaluations of the objective functional $\mathcal{E}$ (up to a constant). We study the well-posedness of the system and its convergence towards minimizers via a mean-field approximation of the dynamics;

iii) We validate the algorithm on various Gaussian VI test problems and investigate the role of different parameters. Comparisons with gradient-based methods demonstrate the superior performance of Gaussian CBO in non-convex low-dimensional settings.

## 2. The Linearized Bures–Wasserstein Geometry

### 2.1. The Bures–Wasserstein manifold

The space of non-degenerate Gaussian measures over $\mathbb{R}^d$ equipped with the BW Riemmanian metric corresponds to the $L^2$-Wasserstein distance (Takatsu, 2011; Malagò et al., 2018; Bhatia et al., 2019).

Let $\mathrm{Sym}_d^{++} \subset \mathrm{Sym}_d$ denote the subset of positive definite symmetric matrices. At each point $(m, \Sigma) \in \mathbb{R}^d \times \mathrm{Sym}_d^{++}$, the tangent space is given by $\mathbb{R}^d \times \mathrm{Sym}_d$ with the usual Euclidean metric in $\mathbb{R}^d$, while $\mathrm{Sym}_d$ is equipped with the scalar product

$$\langle T, S \rangle_\Sigma := \mathrm{tr}(T\Sigma S), \qquad S, T \in \mathrm{Sym}_d.$$

The Riemmanian exponential map is given by

$$\exp_\Sigma(T) := (I + T)\Sigma(I + T) \tag{4}$$

and the logarithmic map by $\log_\Sigma(\overline{\Sigma}) := \overline{\Sigma}(\Sigma\overline{\Sigma})^{-1/2} - I$ (see Appendix A for more details)

A critical aspect of the BW manifold, is that it is not geodesically complete, since the exponential map is well-defined only as long as $I + T$ is positive definitive. In practice this means that starting from $\Sigma$, if we follow a tangent direction $T$, we might hit the boundary of degenerate Gaussian measure where the Riemmanian geometry is not well-defined. Therefore, naively adding noise to the deterministic particle system (2) might result in ill-posed dynamics, as random tangent perturbations may hit the boundary of singular covariance matrices, where the Riemannian geometry degenerates.

We address this issue by linearizing the geometry. This allows us to work in a Hilbert space, where random perturbations are well-defined and barycenters are cheaper to compute.

### 2.2. LBW parametrization

**Linearization.** Linear Optimal Transport (LOT) has recently gained popularity as a computationally efficient way of comparing probability measures while keeping some geometric features of the Wasserstein geometry (Kolouri et al., 2016; Moosmüller & Cloninger, 2023; Sarrazin & Schmitzer, 2024; Cai et al., 2020).

The LOT idea consists of fixing a reference measure $\mu^0 \in \mathcal{P}_2(\mathbb{R}^d)$ and to parametrize every other probability measure $\mu$ with the associated OT map $\mathcal{T}: \mathbb{R}^d \to \mathbb{R}^d$ from $\mu^0$ to $\mu$:

$$\mathcal{T} \mapsto \mu := \mathcal{T}_\# \mu^0.$$

The linearized $L^2$-Wasserstein distance with base $\mu^0$ is then given by

$$\mathrm{LW}_{\mu^0}(\mu^1, \mu^2) := \|\mathcal{T}^1 - \mathcal{T}^2\|_{L^2(\mu^0)}$$

where $\mathcal{T}^i$ is the OT maps associated with $\mu^i \in \mathcal{P}_2(\mathbb{R}^d)$.

We consider the same parametrisation of the space of Gaussian measures, but we also link it the Riemmanian structure of BW. Let $\mu^0 = \mathcal{N}(0, \Sigma^0), \Sigma^0 \in \mathrm{Sym}_d^{++}$ be a reference measure, we parametrize each Gaussian $\mathcal{N}(m, \Sigma)$ with $(m, T), T \in \mathrm{Sym}_d$ where

$$T \mapsto \Sigma = \exp_{\Sigma^0}(T).$$

Due to the domain of definition of $\exp_\Sigma(\cdot)$, we note this parametrization is, to be precise, well-defined only for $T$ such that $T + I \in \mathrm{Sym}_d^{++}$. To avoid such restriction we simply set $\exp_{\Sigma^0}(T) = (I + T)\Sigma^0(I + T)$ for any $T \in \mathrm{Sym}_d$ at the cost of losing the uniqueness of the parametrization. We refer to Appendix B for more details on this technical aspect.

Therefore, instead of defining the particle system in the BW manifold, we will fix a reference measure $\mathcal{N}(0, \Sigma^0)$ and consider a particle system evolving in the tangent space $\mathbb{R}^d \times \mathrm{Sym}_d$ with Linearized BW (LBW) geometry given by

$$\langle (m^1, T^1), (m^2, T^2) \rangle_{\mathsf{LBW}(\Sigma^0)} := \langle m^1, m^2 \rangle + \langle T^1, T^2 \rangle_{\Sigma_0}.$$

**Barycenters.** With respect to BW, we have the benefit of dealing with an unconstrained space, and so without the issue of losing the geometry when hitting the boundary. Also, the computations of barycenters, which we need for the definition of the consensus point, is sensibly cheaper.

Indeed, let $\{(m^i, T^i)\}_{i=1}^N$ be a collection of LBW-parametrized Gaussian probability measures. The associated LBW barycenter with weights $\{\omega^i\}_{i=1}^N, \sum_i \omega^i = 1$ is simply given by $\mathcal{N}(\overline{m}, \overline{\Sigma})$ where $\overline{m} = \sum_i \omega^i m^i$ and

$$\overline{\Sigma} = \exp_{\Sigma^0}(\overline{T}) \quad \text{with} \quad \overline{T} = \sum_{i=1}^N \omega^i T^i. \tag{5}$$

On the contrary, computing BW barycenter would have required solving a matrix equation. See, again, Appendix B for more details, and in particular Figure 6 for a qualitatively comparison between the two notions of barycenters.

**Brownian processes in LBW** To promote exploration in the particle algorithm, we will require to add noise in the dynamics. In LBW, we can conveniently define Brownian processes thanks to the finite-dimensional Hilbert structure given by $\langle \cdot, \cdot \rangle_{\mathsf{LBW}(\Sigma^0)}$.

Let $\{e_\ell\}_{\ell=1}^{d(d+1)/2}$ be an basis for $\mathrm{Sym}_d$ which is orthonormal with respect to $\langle \cdot, \cdot \rangle_{\Sigma^0}$, and $\{\xi_t^\ell\}_{\ell=1}^{d(d+1)/2}$ be independent one-dimensional Brownian processes. A Brownian process $(B_t^{\mathsf{LBW}})_{t \geq 0}$ in LBW is then given by

$$B_t^{\mathsf{LBW}} = (B_t^m, B_t^T) \quad \text{with} \quad B_t^T = \sum_{\ell=1}^{d(d+1)/2} e_\ell \xi_t^\ell,$$

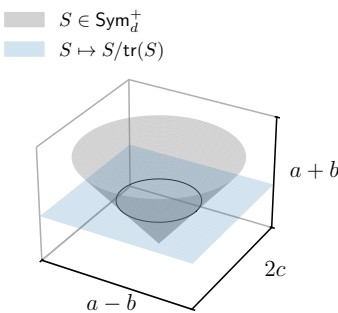

*Figure 2.* To visualize $S \in \mathsf{Sym}_d^+$, $S = (a, c; c, b)$, we first map it to $\mathbb{R}^3$ as $(a - b, 2c, a + b)$. The cone $\mathsf{Sym}_d^+$ is then given by $z \geq \sqrt{x^2 + y^2}$. The 2D plot is finally obtained by projection towards trace 1 matrices $S \mapsto S/\mathrm{tr}(S)$ (via $(x, y, z) \mapsto (x, y)/z$).

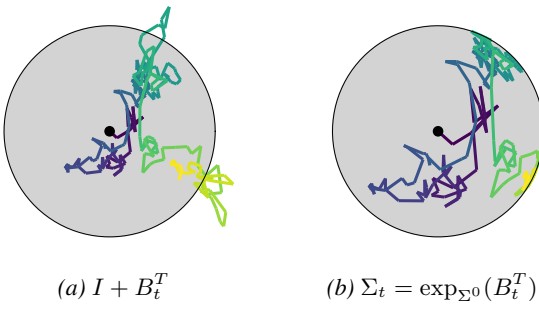

(a) $I + B_t^T$    (b) $\Sigma_t = \exp_{\Sigma^0}(B_t^T)$

*Figure 3.* A Brownian path in LBW space. If $B_t^T$ is a Brownian particle in $\mathsf{Sym}_d$, it may leave the cone of OT maps, see (a) (equivalently, $I + B_t^T$ leaves the cone of positive semi-definite matrices $\mathsf{Sym}_d^+$). The extended exponential map (4), though, automatically reflects the dynamics so that $\Sigma_t = \exp_{\Sigma^0}(B_t^T) \in \mathsf{Sym}_d^+$ without additional computational effort, see (b).

where $B_t^m$ is a $d$-dimensional Brownian process.

Remarkably, the associated Gaussian measure is a random process in $\mathcal{N}_d$ which may reach the boundary of degenerate Gaussian measures but the dynamics remains well-defined, see Figures 2 and 3 for an intuition in the $d = 2$ case.

## 3. Gaussian consensus-based optimization

### 3.1. The particle dynamics

We are now ready to define the Consensus-Based Optimization (CBO) particle system in the LBW space. The $N$ Gaussian particles at time $t \geq 0$ are described tangent vectors at the reference measures $\mu^0 = \mathcal{N}(0, \Sigma^0)$, $\Sigma^0 \in \mathsf{Sym}_d^{++}$

$$(m_t^i, T_t^i) \in \mathbb{R}^d \times \mathsf{Sym}_d \qquad i = 1, \ldots, N$$

via the relation $\mu_t^i = \mathcal{N}\left(m_t^i, \Sigma_t^i\right)$, with $\Sigma_t = \exp_{\Sigma^0}(T_t^i)$.

A CBO-type dynamics (Pinnau et al., 2017) is characterized by a deterministic component which drives particles towards the consensus point and a stochastic component favoring exploration of the search space.

**Consensus point.** As in the deterministic consensus dynamics (2), the consensus point corresponds to a weighted barycenter, now computed according to the LBW geometry. To stress the the dependence of the barycenter on the entire particle system, we introduce the empirical measure $\rho_t^N = (1/N) \sum_{i=1}^{N} \delta_{(m_t^i, T_t^i)} \in \mathcal{P}(\mathbb{R}^d \times \mathsf{Sym}_d)$.

The consensus point is then given by the weighted LBW barycenter (5) with exponential weights (3). For an arbitrary measure $\rho \in \mathcal{P}(\mathbb{R}^d \times \mathsf{Sym}_d)$, it reads

$$\overline{m}^\alpha[\rho] := \frac{\int m\, e^{-\alpha \mathcal{E}^{\#}(m, T)} \rho(\mathrm{d}m, \mathrm{d}T)}{\int e^{-\alpha \mathcal{E}^{\#}(m, T)} \rho(\mathrm{d}m, \mathrm{d}T)},$$
$$\overline{T}^\alpha[\rho] := \frac{\int T\, e^{-\alpha \mathcal{E}^{\#}(m, T)} \rho(\mathrm{d}m, \mathrm{d}T)}{\int e^{-\alpha \mathcal{E}^{\#}(m, T)} \rho(\mathrm{d}m, \mathrm{d}T)} \tag{6}$$

where, for notational simplicity, we introduced the finite-dimensional objective function

$$\mathcal{E}^{\#}(m, T) := \mathcal{E}(\mathcal{N}(m, \exp_{\Sigma^0}(T))).$$

For the empirical measure $\rho_t^N$, the consensus point $(\overline{m}^\alpha[\rho_t^N], \overline{T}^\alpha[\rho_t^N])$ reduces to a weighted sum of the particles, which, for $\alpha \gg 1$, is close to the best particles among the ensemble thanks to the Boltzmann–Gibbs exponential weights. Therefore, the consensus point can be considered a proxy for the best particle of the ensemble.

**Evolution.** Given two vectors $a, b \in \mathbb{R}^d$, we denote with $a \odot b \in \mathbb{R}^d$ the component-wise product, $(a \odot b)_\ell = a_\ell b_\ell$. We fix a basis $\boldsymbol{e} = \{e_\ell\}_{\ell=1}^{d(d+1)/2}$ for $\mathsf{Sym}_d$ orthonormal with the respect to $\langle \cdot, \cdot \rangle_{\Sigma^0}$. For $S, T \in \mathsf{Sym}_d$, we define the component-wise product as

$$T \odot S := \left(\sum_\ell T_\ell^{\boldsymbol{e}} e_\ell\right) \odot \left(\sum_\ell S_\ell^{\boldsymbol{e}} e_\ell\right) = \sum_\ell T_\ell^{\boldsymbol{e}} S_\ell^{\boldsymbol{e}} e_\ell$$

which corresponds to the component-wise product between the vectors of coefficients $(T_1^{\boldsymbol{e}}, \ldots, T_{d(d+1)/2}^{\boldsymbol{e}})$, $(S_1^{\boldsymbol{e}}, \ldots, S_{d(d+1)/2}^{\boldsymbol{e}})$.

Let $B_t^i = (B_t^{i,m}, B_t^{i,T})$ be $N$ independent Brownian processes taking values in $\mathbb{R}^d \times \mathsf{Sym}_d$ constructed with the basis $\boldsymbol{e}$. The CBO dynamics in LBW space reads for $i \in [N]$

$$\begin{cases} \mathrm{d}m_t^i = \lambda(\overline{m}^\alpha[\rho_t^N] - m_t^i)\mathrm{d}t \\ \qquad\quad + \sigma(\overline{m}^\alpha[\rho_t^N] - m_t^i) \odot \mathrm{d}B_t^{i,m} \\ \mathrm{d}T_t^i = \lambda(\overline{T}^\alpha[\rho_t^N] - T_t^i)\mathrm{d}t \\ \qquad\quad + \sigma(\overline{T}^\alpha[\rho_t^N] - T_t^i) \odot \mathrm{d}B_t^{i,T} \end{cases} \tag{7}$$

with $(m_0^i, T_0^i) \sim \rho_0$, i.i.d, for some $\rho_0 \in \mathcal{P}(\mathbb{R}^d \times \mathsf{Sym}_d)$.

Above, $\lambda, \sigma > 0$ are two parameters controlling the strength of the deterministic and stochastic components respectively. We note that noise is possibly degenerate as it depends on the differences $(\overline{m}^\alpha[\rho_t^N] - m_t^i)$ and $(\overline{T}^\alpha[\rho_t^N] - T_t^i)$. In this way, particles which are far from the consensus point tend to have a more explorative behavior than those close to it.

This is an essential mechanism for ensuring emergence of consensus as the particles evolve. Also, the diffusion is anisotropic since each direction is explored at a different rate. This strategy has been proposed in (Carrillo et al., 2021) for superior performance in high-dimensional problems.

It is important to remark that, thanks to (6)-(7) and the definition of $\odot$, the CBO dynamics in LBW corresponds to the standard Euclidean CBO particle system if we identify each $T_t^i$ with its coefficients associated with the basis $e = \{e_\ell\}_{\ell=1}^{d(d+1)/2}$. Therefore, (7) corresponds to a CBO dynamics in $\mathbb{R}^D$ with $D = d + d(d+1)/2$, and we can rely on the standard well-posedness results for CBO particle systems (Carrillo et al., 2021; 2018).

## 3.2. Well-posedness

For completeness, we recall in the following the well-posedness results and translate the assumption on the (finite-dimensional) objective function $\mathcal{E}^\# = \mathcal{E}^\#(m, T)$ into assumptions on the functional $\mathcal{E} = \mathcal{E}(\mu)$ for better interpretability. The proof of each result is collected in Appendix C.1. Recall $\mathbb{LW}_{\mu^0}$ is the LOT distance, while we denote with simply $\mathbb{W}$ the $L^2$-Wasserstein distance.

**Lemma 3.1.** *Let $\Sigma^0 \in \mathrm{Sym}_d^{++}$ and $\mu^0 = \mathcal{N}(0, \Sigma^0)$. Assume $\mathbb{E}|m_0^i|^2, \mathbb{E}\|T_0^i\|_{\Sigma^0}^2 < 0$, and that, for some $L_\mathcal{E} > 0$ it holds for any $\mu^1, \mu^2 \in \mathcal{P}_2(\mathbb{R}^d)$*

$$|\mathcal{E}(\mu^1) - \mathcal{E}(\mu^2)| \leq L_\mathcal{E} \left(1 + M_2(\mu^1) + M_2(\mu^2)\right) \times \mathbb{LW}_{\mu^0}(\mu^1, \mu^2). \quad (8)$$

*Then, system (7) admits a unique strong solution.*

Let us discuss what type of functional $\mathcal{E}$ satisfy condition (8). For energy functionals of type

$$\mathcal{V}(\mu) = \int V(x)\mu(\mathrm{d}x), \quad \mathcal{W}(\mu) = \iint W(x, y)\mu(\mathrm{d}x)\mu(\mathrm{d}y) \quad (9)$$

a sufficient condition for (8) is the local Lipschitz continuity of $V$ and $W$. More delicate is the case of the log-entropy (relevant for the KL divergence)

$$\mathcal{U}(\mu) = \begin{cases} \int \log(\mu(x))\mu(\mathrm{d}x) & \text{if } \mu \ll \mathcal{L}^d \\ +\infty & \text{otherwise} \end{cases}$$

as it takes the infinite value for singular Gaussian measures. Though, $\mathcal{U}$ satisfies a local Lipschitz bound for the $L^2$-Wasserstein distance $\mathbb{W}$ (which is stronger than a bound for

$\mathbb{LW}_{\mu^0}$) under a regularity condition (see (Polyanskiy & Wu, 2016), Proposition 1, recalled in Appendix C)

Let $\mathrm{clip}_\varepsilon : \mathrm{Sym}_d^+ \to \{\Sigma \in \mathrm{Sym}_d^{++} : \Sigma \succcurlyeq \varepsilon I\}$ be the function which clips the eigenvalues to a minimum value $\varepsilon > 0$ defined by $\mathrm{clip}_\varepsilon(\Sigma) := \sum_\ell \max\{\lambda_\ell, \varepsilon\} u_\ell u_\ell^\top$ for $\Sigma = \sum_\ell \lambda_\ell u_\ell u_\ell^\top$, where $(\lambda_\ell, u_\ell)_\ell$ is an eigenbasis for $\Sigma$, as in (Lambert et al., 2022). Then, we may now regularize the entropy functional by setting for $\mu = \mathcal{N}(m, \Sigma)$

$$\mathcal{U}_\varepsilon(\mu) := \mathcal{U}\left(\mathcal{N}(m, \mathrm{clip}_\varepsilon(\Sigma))\right). \quad (10)$$

This modification does not affect the location of the minimizers for $\varepsilon \ll 1$ and ensures well-posedness of the particle system (7). We also have the following result.

**Lemma 3.2.** *Let $\mu^{\mathrm{targ}} \propto \exp(-V)$, with $V$ such that $|V(x) - V(y)| \leq L_V(1 + |x| + |y|)|x - y|$ for any $x, y \in \mathbb{R}^d$.*

*The regularized divergence $\mathrm{KL}_\varepsilon(\cdot|\mu^{\mathrm{targ}}) : \mathcal{N}^d \mapsto [0, \infty)$,*

$$\mathrm{KL}_\varepsilon(\mu|\mu^{\mathrm{targ}}) = \mathcal{V}(\mu) + \mathcal{U}_\varepsilon(\mu)$$

*satisfies the Lipschitz condition (8).*

We remark that the clipping is only necessary for the well-posedness of the time-continuous dynamics, and that, in practice, one can implement the algorithm by simply truncating the value of $\mathcal{E}$.

More general internal energies can also be considered provided they satisfy (8). For instance, in Appendix C.1 we also study the case of Maximum Mean Discrepancy (MMD).

## 3.3. Mean-field analysis

Mean-field approximations of interacting particle systems are a powerful tool to study their long time behavior. For CBO methods, the mean-field analysis allows to investigate the effectiveness of the algorithm by studying its convergence towards global minimizers (Carrillo et al., 2018; 2021; Fornasier et al., 2024).

We show now that the same type of analysis also applies to the Gaussian CBO particle system (7). As for Lemma 3.1, we can rely entirely on the results available in the literature for standard CBO in $\mathbb{R}^D$ thanks to the finite-dimensional and Euclidean nature of $\mathbb{R}^d \times \mathrm{Sym}_d$ equipped with the LBW product. We translate, when possible, the assumptions on $\mathcal{E}^\#$ into assumptions on $\mathcal{E}$.

**Chaos propagation.** The mean-field approximation of the particle system (7) can be formally derived by assuming propagation of chaos of the particle system (Sznitman, 1991). Let $F_t \in \mathcal{P}\left((\mathbb{R}^d \times \mathrm{Sym}_d)^N\right)$ be the particles joint probability measure. If $F_0 = \rho_0^{\otimes N}$, we assume that for large particle systems $N \gg 1$ the distribution at $t \geq 0$ can be approximated as $F_t \approx \rho_t^{\otimes N}$ for some $\rho_t \in \mathcal{P}(\mathbb{R}^d \times \mathrm{Sym}_d)$.

This means that the particles are i.i.d also at subsequent times $t \geq 0$, and that each of them evolves according to the McKean–Vlasov process

$$\begin{cases} \mathrm{d}m_t = \lambda(\overline{m}^\alpha[\rho_t] - m_t)\mathrm{d}t + \sigma(\overline{m}^\alpha[\rho_t] - m_t) \odot \mathrm{d}B_t^m \\ \mathrm{d}T_t = \lambda(\overline{T}^\alpha[\rho_t] - T_t)\mathrm{d}t + \sigma(\overline{T}^\alpha[\rho_t] - T_t) \odot \mathrm{d}B_t^T \\ \rho_t = \mathrm{Law}(m_t, T_t) \,. \end{cases}$$

$$(11)$$

Note that the average mean $\overline{m}^\alpha[\rho_t^N]$ is substituted above by $\overline{m}^\alpha[\rho_t]$, which depends on the own particle law $\rho_t$.

**Well-posedness.** We refer to Appendix C.2 for the proofs.

**Assumption 3.3.** The objective functional $\mathcal{E}$ is bounded from below over $\mathcal{N}^d$, $\underline{\mathcal{E}} := \inf_{\mu \in \mathcal{N}^d} \mathcal{E}(\mu)$ and locally Lipschitz continuous (8). Furthermore, either $\mathcal{E}$ is bounded from above, $\sup_{\mu \in \mathcal{N}^d} \mathcal{E}(\mu) < \infty$, or it grows quadratically at infinity:

$$\mathcal{E}(\mu) - \underline{\mathcal{E}} \geq c_l M_2(\mu)^2 \qquad \text{for} \quad \mu, \ M_2(\mu) \geq R \,,$$

for some constants $R, c_l > 0$.

**Lemma 3.4.** *Let* $\mu^0 = \mathcal{N}(0, \Sigma^0), \Sigma^0 \in \mathrm{Sym}_d^{++}$, $\mathcal{E}$ *satisfy Assumption 3.3, and* $\rho_0 \in \mathcal{P}_4(\mathbb{R}^d \times \mathrm{Sym}_d)$. *Then, there exists a unique non-linear process* $(m, T) \in \mathcal{C}([0, \infty), \mathbb{R}^d \times \mathrm{Sym}_d)$ *satisfying* (11) *in a strong sense with* $\lim_{t \to 0} \rho_t = \rho_0 \in \mathcal{P}_2(\mathbb{R}^d \times \mathrm{Sym}_d)$.

We have already discussed under which conditions the functionals $\mathcal{V}, \mathcal{W}, \mathcal{U}$ satisfy the local Lipschitz continuity (8). Lower bound for $\mathcal{V}, \mathcal{W}$ follows from lower bound the $V$ and $W$ respectively. Clearly, if $\inf V, \inf W > -\infty$, then $\inf \mathcal{V}, \inf \mathcal{W} > -\infty$. Also, if $V, W$ grow quadratically at infinity, so does $\mathcal{V}, \mathcal{W}$ (see Lemma C.3).

For $\mathcal{U}$, we cannot expect quadratic growth at infinity, nor boundedness. Therefore, only the bounded, clipped version $\mathcal{U}_\varepsilon$ (10) satisfies Assumption 3.3 and, as a consequence, the same holds for the regularized divergence $\mathrm{KL}_\varepsilon$ from a target $\mu^{\mathrm{targ}} \propto \exp(-V)$.

In the case of CBO-type dynamics, the propagation of chaos assumption can be actually substituted by rigorous mean-field limit results. This justifies the study the model (11) to understand the algorithm convergence properties, see (Gerber et al., 2025) for more details and updated references.

### 3.4. Convergence towards global minima

For $\alpha > 0$, recall the particle weights are given by $\omega(\mu) = \exp(-\alpha\mathcal{E}(\mu))$, and let us consider the corresponding finite-dimensional ones $\omega^\#(z) := \exp(-\alpha\mathcal{E}^\#(z))$ for $z \in \mathbb{R}^d \times \mathrm{Sym}_d$. The cornerstone of the convergence analysis of CBO methods is the Laplace principle (Dembo & Zeitouni, 2010) which states that for a compactly supported $\rho \in \mathcal{P}(\mathbb{R}^d \times$

$\mathrm{Sym}_d)$, it holds

$$\lim_{\alpha \to \infty} -\frac{1}{\alpha} \log \int \exp(-\alpha\mathcal{E}^\#(z))\rho(\mathrm{d}z) = \inf_{z \in \mathrm{supp}(\rho)} \mathcal{E}^\#(z) \,.$$

$$(12)$$

For a quantitative version in terms of consensus points, see Proposition 4.5 in (Fornasier et al., 2024). In the following, we denote with $\mathrm{Var}(\rho)$ the variance of a probability measure $\rho$: $\mathrm{Var}(\rho) = (1/2) \int \|z_1 - z_1\|^2 \rho(\mathrm{d}z_1)\rho(\mathrm{d}z_2)$

We recall the convergence result from (Carrillo et al., 2021) (Assumption 3.1, Theorem 3.2) applied to the finite dimensional setting in the space $\mathbb{R}^d \times \mathrm{Sym}_d$.

**Theorem 3.5.** *Assume* $\underline{\mathcal{E}} := \inf \mathcal{E}^\# > -\infty$, $\mathcal{E}^\# \in \mathcal{C}^2(\mathbb{R}^d \times \mathrm{Sym}_d)$ *with bounded second derivatives, that is, for an orthonormal basis* $\{e_\ell\}_{\ell=1}^D$, $D = d + d(d+1)/2$ *there exists* $c_\mathcal{E}$ *such that* $\max_\ell \max_z |\partial^2 \mathcal{E}^\#(z)/\partial e_\ell^2| < c_\mathcal{E}$.

*If* $\alpha, \lambda, \sigma$ *and the initial distribution* $\rho_0$ *is chosen such that* $\mathrm{argmin}\,\mathcal{E}^\#(\mu) \subset \mathrm{supp}(\rho_0)$ *and*

$$C_1 := 2\lambda - \sigma^2 - 2\sigma^2 \frac{e^{-\alpha\underline{\mathcal{E}}}}{\|\omega^\#\|_{L^2(\rho_0)}} > 0 \,,$$

$$C_2 := \frac{2\mathrm{Var}(\rho_0)}{C_1 \|\omega^\#\|_{L^2(\rho_0)}} \alpha e^{-\alpha\underline{\mathcal{E}}} c_\mathcal{E}(2\lambda + \sigma^2) \leq \frac{3}{4} \,,$$

*then* $\mathrm{Var}(\rho_t) \to 0$ *exponentially fast and there exists* $\tilde{z}$ *such that the consensus point and* $\int z\rho_t(\mathrm{d}z)$ *converge to* $\tilde{z}$ *exponentially fast. Moreover, it holds that*

$$\mathcal{E}^\#(\tilde{z}) \leq \underline{\mathcal{E}} + r(\alpha) + \frac{\log 2}{\alpha} \,,$$

*where* $r(\alpha) := -(1/\alpha) \log \|\omega^\#\|_{L^2(\rho_0)} - \underline{\mathcal{E}} \to 0$ *as* $\alpha \to \infty$ *by the Laplace principle* (12).

*Remark* 3.6. We note that parameters $\lambda, \sigma, \alpha$ and $\rho_0$ can always be picked to satisfy the Theorem's assumption by choosing, for $C_1$, $\sigma$ sufficiently small, and for $C_2$, $\mathrm{Var}(\rho_0)$ sufficiently small. The two key aspects in the assumptions are that $\sigma^2 \lesssim 2\lambda$ and that $\mathrm{Var}(\rho_0)$ should be small. The first condition comes from an intrinsic property of the particle dynamics and says that, for consensus emergence to occur, the diffusion parameter $\sigma$ should not be too large.

The requirement that $\mathrm{Var}(\rho_0)$ is small, instead, is more related to the variance-based proof strategy than to the particle dynamics itself. Indeed, such a restriction is not present when using the different analysis strategy proposed in (Fornasier et al., 2024) (see Remark C.4 for more details). Finally, we remark that the result holds at the mean-field level, and that the accuracy of the mean-field approximation typically deteriorates for $\alpha \gg 1$ (Gerber et al., 2025), showing a trade-off between accuracy and computational cost of the algorithm.

For $\mathcal{V}$ and $\mathcal{W}$, differentiability with respect to the mean follows directly from that of $V$ and $W$. Precisely, we have

---

**Algorithm 1** Gaussian Consensus-Based Optimization

---

**Input:** Objective function $\mathcal{E}$, reference $\mu^0 = \mathcal{N}(0, \Sigma^0)$
  parameters $\lambda = 1, \sigma, \Delta t > 0, \alpha \gg 1, N \in \mathbb{N}$
Initialize particles $(m^i, T^i) \in \mathbb{R}^d \times \mathrm{Sym}_d, i \in [N]$
Evaluate objective $\mathcal{E}(\mu^i)$ with $\mu^i = \mathcal{N}(m^i, \exp_{\Sigma^0}(T^i))$
**repeat**
  Set particle weights $\omega^i \propto \exp(-\alpha \mathcal{E}(\mu^i))$
  Compute consensus $(\overline{m}^\alpha, \overline{T}^\alpha)$ with $\{\omega^i\}_{i=1}^N$
  **for (parallel)** $i = 1$ **to** $N$ **do**
    Sample random normal vectors $(B^{i,m}, B^{i,T})$
    Update particle $(m^i, T^i)$ with (13)
    Update objective value $\mathcal{E}(\mu^i)$
  **end for (parallel)**
**until** convergence reached
**Output:** Consensus Gaussian $\mathcal{N}(\overline{m}^\alpha, \exp_{\Sigma^0}(\overline{T}^\alpha))$

---

that the assumptions are verified provided $W \in \mathcal{C}^4(\mathbb{R}^d \times \mathbb{R}^d)$, with bounded second- and fourth-order derivatives (see Appendix C.3).

The log-entropy $\mathcal{U}$ is invariant under mean shifts, so $\nabla_m \mathcal{U}(\mu) = 0$. For $\Sigma \in \mathrm{Sym}_d^{++}$, see Appendix A.1 in (Lambert et al., 2022), it holds $\nabla_\Sigma \mathcal{U}(\mu) = -(1/2)\Sigma^{-1}$. Moreover, let $e_\ell$ be the $\ell$-th canonical basis vector of $\mathbb{R}^d$, we have, see (Giles, 2008),

$$\frac{\partial}{\partial \Sigma_{ij}} \Sigma^{-1} = -\Sigma^{-1} e_i e_j^\top \Sigma^{-1}.$$

Therefore, the boundedness assumptions in Theorem 3.5 on the second derivatives, hold only on if we stay far away from singular Gaussian measures, in a subset $\Sigma \succ \varepsilon I, \varepsilon > 0$.

In practice, applying a smooth eigenvalue-clipping procedure to $\Sigma$, as done in (10), ensures the covariance remains in this admissible set, which in turn guarantees convergence towards global minimizers. This regularization affects the objective functional only near singular covariance matrices. Therefore, if the minimizer of the original energy has a non-degenerate covariance, the regularization should not affect the optimization problem. A detailed analysis of such regularization is beyond the scope of this work.

## 4. Experiments

### 4.1. Algorithm

We discuss now the implementation aspects of the Gaussian CBO dynamics and validate its optimization capabilities against different test problems. Further details on the experimental settings can be found in Appendix D, and see Algorithm 1 for a pseudocode description. The code used for the experiments is available at https://github.com/borghig/GaussCBO.

**Time discretization.** Let $\mu^0 = \mathcal{N}(0, \Sigma^0), \Sigma^0 \in \mathrm{Sym}_d^{++}$ be a reference measure. We discretize the CBO particle dynamics (7) via a simple Euler–Maruyama scheme with step size $\Delta t > 0$:

$$\begin{cases} m_{(k+1)}^i = m_{(k)}^i + \Delta t\, \lambda(\overline{m}^\alpha[\rho_{(k)}^N] - m_{(k)}^i) \\ \qquad\quad + \sqrt{\Delta t}\, \sigma(\overline{m}^\alpha[\rho_{(k)}^N] - m_{(k)}^i) \odot B_{(k)}^{i,m}, \\ T_{(k+1)}^i = T_{(k)}^i + \Delta t\, \lambda(\overline{T}^\alpha[\rho_{(k)}^N] - T_{(k)}^i) \\ \qquad\quad + \sqrt{\Delta t}\, \sigma(\overline{T}^\alpha[\rho_{(k)}^N] - T_{(k)}^i) \odot B_{(k)}^{i,T}, \end{cases} \quad (13)$$

for $i \in [N]$. Here, $B_{(k)}^{i,m} \sim \mathcal{N}(0, I)$ are i.i.d., while $B_{(k)}^{i,T}$ are standard normal vectors with respect to the scalar product $\langle \cdot, \cdot \rangle_{\Sigma^0}$ (in Appendix D.1 we show how to sample them).

**Baseline.** We compare the results with different methods based on the evolution of a Gaussian-measure according to different geometries: Bures–Wasserstein (BW) gradient flow (Lambert et al., 2022), Gaussian Stein Variational Gradient Descent (SVGD) with kernel $K_1(x, y) = x^\top y + 1$ (Liu et al., 2023), and (natural) Fisher–Rao (FR) gradient flow (Barfoot, 2020). Details of the methods can be found in Appendix D.4. The objective function is $\mathrm{KL}(\mu \,|\, \mu^{\mathrm{targ}})$ for all methods, and the flows are discretized via an explicit Euler scheme with step size $\Delta t$ and the same quadrature approximation for the expected values.

**Settings.** In our experiments, the objective $\mathcal{E}$ is the KL divergence from the target measure $\mu^{\mathrm{targ}} \propto \exp(-V)$. Recall that, to compute the consensus point, we need to evaluate the functional $\mathcal{E}$ at the particle locations. Since $\mathrm{KL}(\mu \,|\, \mu^{\mathrm{targ}}) = \mathcal{U}(\mu) + \mathcal{V}(\mu)$, we set $\mathcal{E}(\mu) = M = 10^4$ whenever the particle's covariance matrix is singular (when $\mathcal{U}(\mu) = +\infty$). The expected value $\mathcal{V} = \int V(x)\mu(\mathrm{d}x)$, $V := -\log(\mu^{\mathrm{targ}})$ is approximated using a quadrature rule based on $2d + 1$ points (Arasaratnam & Haykin, 2009), although a Monte Carlo estimate could also be used.

### 4.2. Tests

**Case** $d = 2$. We test the algorithm for different Gaussian Variational Inference (VI) problems where targets are Gaussian mixture models with $K = 2$ or $K = 4$ components

$$\mu^{\mathrm{targ}} = \sum_{k=1}^K w_k \mathcal{N}(m_k, \Sigma_k) \qquad (14)$$

see Table 1 in Appendix D.2 for the exact definitions.

Since system (13) is over-parameterized, in CBO algorithms one typically fixes $\lambda = 1$ (Carrillo et al., 2021). The other parameters are set to $\Delta t = 0.05$, $\sigma = 5$, $\alpha = 10^4$, and $N = 20$. We keep $\Sigma^0 = I$ to be the reference measure throughout the computation. Given an initialization

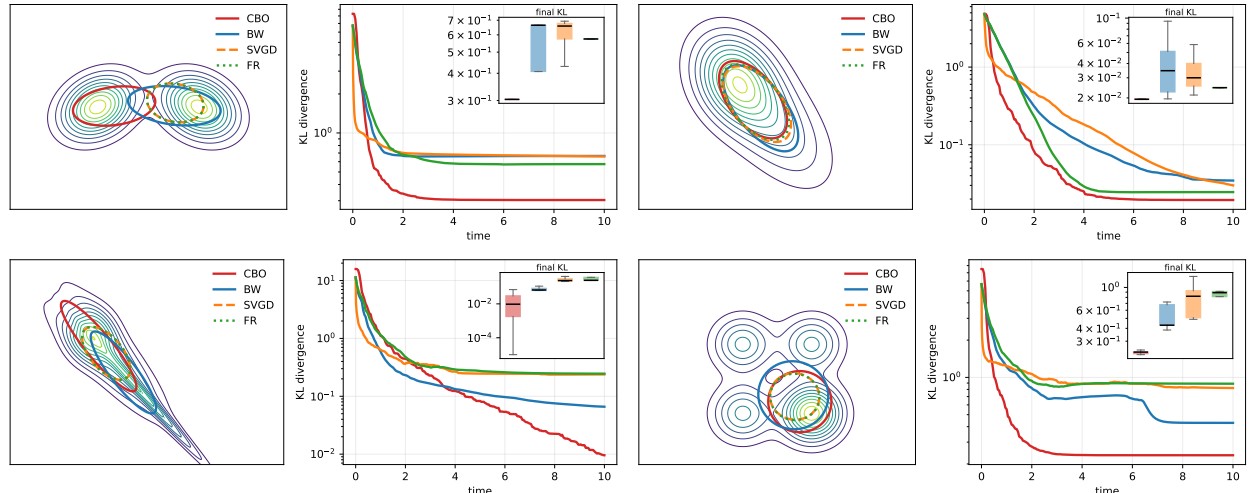

*Figure 4.* Comparison between CBO and baseline algorithms in approximating different target densities given by Gaussian mixture models with 2 components (A, B) and 4 components (C, D) (see Table 1). The 2D plots show the computed solutions for a single run, while the KL evolution is averaged over 100 runs with different random initializations. Median and $[0.25, 0.75]$ interquartile ranges are shown. Parameters: $\Delta t = 0.05$ (except for SVGD, $\Delta t/4$, and FR $\Delta t/2$, for better stability),, $\lambda = 1, \sigma = 5, N = 20, \alpha = 10^4$. Reference measure for LBW is $\mathcal{N}(0, I)$. Supplementary videos illustrating the evolution are available here (Borghi & Carrillo, 2025).

$(m_{(0)}, \Sigma_{(0)})$ for the baseline algorithms, the CBO particles are initialized as $m^i_{(0)} = m_{(0)} + 0.1\xi^i, \xi^i \sim \mathcal{N}(0, I)$, and $T^i_{(0)} = \log_I(\Sigma_{(0)}) + 0.1\Xi^i$, where $\Xi^i_{\ell k} \sim \mathcal{N}(0, 1)$ and $\Xi^i = (\Xi^i)^\top$. This makes the comparison between baseline and CBO fairer, as it prevents the CBO particles from exploring the search space even before the dynamics begins.

We perform the same experiments for different target measures (A, B, C, and D) and collect statistics over 100 runs. The starting points $(m_{(0)}, \Sigma_{(0)})$ are initialized as $m_{(0)} \sim \text{Unif}([-5, 5]^2)$ and $\Sigma_{(0)} = I$ in all runs. For all targets, Figure 4 shows one illustrative run and the median KL divergence evolution over 100 runs, together with the $[0.25, 0.75]$ interquartile range. CBO outperforms baseline algorithms in all scenarios considered: not only for non-logconcave targets (tests A, D) but also for unimodal ones (tests B, C). For the baseline algorithms SVGD and FR, the time step is reduced to $\Delta t/4$ and $\Delta t/2$, respectively, to avoid numerical instability in the Euler discretization.

**Algorithmic sensitivity.** The sensitivity of Gaussian CBO with respect to key algorithmic parameters, namely the diffusion strength $\sigma$, the number of particles $N$, and the choice of reference measure for the linearization of the LBW geometry, is analyzed in Appendix D.2.

Performance is influenced by $\sigma$, with intermediate values providing the best balance between exploration and consensus, whereas increasing the number of particles beyond a modest threshold yields only marginal gains. The effect of updating the reference Gaussian during the optimization to re-center the linearization around the current consensus

was also examined, but no systematic improvement was observed in the considered problems.

**Case** $d = 10$. We also assess the performance of the baseline methods and Gaussian CBO on synthetic Gaussian mixture targets in dimension $d = 10$. A detailed description of the experimental setup, normalization procedure, and parameter choices is provided in Appendix D.3. Figure 5 reports the aggregated evolution of the relative KL divergence across multiple random instances. Overall, Gaussian CBO exhibits more stable behavior than BW and FR baseline with a smaller interquartile range and better average performance. On average, though, CBO exhibits a worse performance than the SVGD baseline algorithm. For stability, time-steps of FR and SVGD were reduced to $\Delta t/20$.

**Limitations.** The proposed method requires multiple function evaluations due to the particle-based nature of the optimizer. While these evaluations can be performed in parallel, this may limit its applicability in high-dimensional settings. As mentioned, the tuning of the noise parameter $\sigma$ is also delicate and might be dimension-dependent (Carrillo et al., 2021). Moreover, in high dimension, the number of covariance parameters scales like $d^2$, so covariance optimization may dominate the mean optimization. A possible solution is to restrict the LBW geometry to Gaussians with diagonal covariances, as in (Petit-Talamon et al., 2026).

Finally, the current approach returns a single Gaussian approximating the target measure, which might not be suitable for complex VI problems. In Appendix E, we discuss a possible extension to Gaussian mixture models (GMMs)

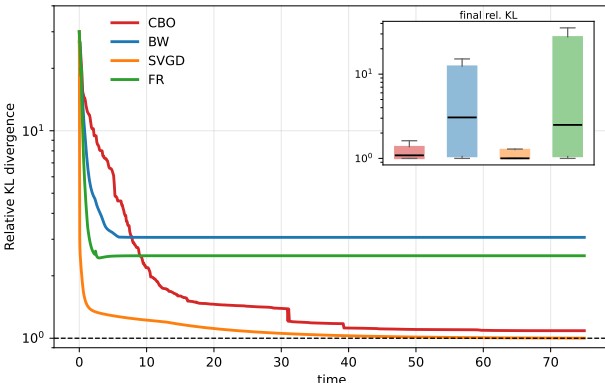

*Figure 5.* Comparison between CBO and baselines in approximating Gaussian mixture targets ($K = 5$) in $d = 10$. The curves report the relative KL divergence $\mathrm{RelKL}(t) = \mathrm{KL}(t)/K^\star$, normalized by the best value achieved on each instance, and averaged across $M = 20$ random mixtures. Median and $[0.25, 0.75]$ interquartile ranges are shown. Parameters: $\Delta t = 0.1$, $T = 75$, $\lambda = 1$, $\sigma = 2.5$, $N = 100$, $\alpha = 10^4$. For SVGD and FR, the time steps are reduced to $\Delta t/20$ for better stability. Base measure not updated.

based on the introduction of multiple swarms of particles.

*Remark* 4.1. The baseline algorithms are deterministic methods, while in the proposed CBO algorithm particles randomly explore the search space via Brownian paths. In the literature, stochastic versions of first-order optimizers have been proposed; see, for instance, (Lambert et al., 2022), where the randomness comes from Monte Carlo evaluations of the objective functional rather than from explicit exploratory components in the dynamics. These are two different and possibly complementary notions of stochasticity, and we leave a systematic comparison for future work.

## 5. Outlook

We have introduced a new computational paradigm based on Gaussian particles evolving according to a stochastic CBO-type dynamics in the Linearized Bures–Wasserstein (LBW) space.

The proposed framework enables efficient simulation of interacting particle systems while preserving essential features of the underlying Bures–Wasserstein geometry. Beyond the specific algorithm developed here, the LBW representation opens the door to a variety of new stochastic optimization dynamics on the space of Gaussian measures.

It is also natural to ask whether one can define and simulate the particle dynamics directly in the full Bures–Wasserstein space, without relying on linearization. This would require a careful treatment of singular Gaussians, which is nontrivial in the non-linear geometry of $\mathcal{N}^d$.

Finally, the optimal transport viewpoint also opens the door

to extensions of particle-based optimizers beyond the Gaussian setting, for instance along the lines of Linear Optimal Transport. This would allow to handle broader classes of probability measures while retaining the consensus-based optimization paradighm.

## Aknowledgments

GB was supported by the Wolfson Fellowship of the Royal Society "Uncertainty quantification, data-driven simulations and learning of multiscale complex systems governed by PDEs" of Prof. L. Pareschi at Heriot-Watt University. JAC was supported by the Advanced Grant Nonlocal-CPD (Nonlocal PDEs for Complex Particle dynamics: Phase Transitions, Patterns and Synchronization) of the European Research Council Executive Agency (ERC) under the European Union's Horizon 2020 research and innovation programme (grant agreement No. 883363). JAC was also partially supported by the "Maria de Maeztu" Excellence Unit IMAG, reference CEX2020-001105-M, funded by the Spanish ministry of Science MCIN/AEI/10.13039/501100011033/ and the EPSRC grant numbers EP/T022132/1 and EP/V051121/1.

## Impact Statement

This paper presents work whose goal is to advance the field of optimization and there are not potential societal consequences of our work.

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

# A. Background on the Bures–Wasserstein space

In this section we recall in detail the geometry of the Bures–Wasserstein manifold as its relation with the $L^2$-Wasserstein distance between arbitrary probability measures.

## A.1. Notation

The set $\mathcal{P}(\mathbb{R}^d)$ is the set of Borel probability measure over $\mathbb{R}^d$, and $\mathcal{P}_2(\mathbb{R}^d)$ is the subset of measures with bounded second moments: $\mu \in \mathcal{P}(\mathbb{R}^d)$ with $M_2(\mu) := (\int |x|^2 \mu(\mathrm{d}x))^{1/2} < \infty$. $\mathcal{P}_2^{ac} \subset \mathcal{P}(\mathbb{R}^d)$ is the subset of measures admitting a density with respect to Lebesgue and we will sometimes abuse the notation by denoting the density of $\mu$ with $\mu = \mu(x)$ itself. For a point $x \in \mathbb{R}^d$, $\delta_x \in \mathcal{P}(\mathbb{R}^d)$ is the Dirac measure: $\delta_x(A) = 1$ if $x \in A$, and 0 otherwise.

With $\mathrm{Sym}_d$ we indicate the set of $d \times d$ symmetric matrices, while $\mathrm{Sym}_d^+, \mathrm{Sym}_d^{++} \subset \mathrm{Sym}_d$ are, respectively, the sets of positive semi-definite and positive definite symmetric matrices. Given a mean $m \in \mathbb{R}^d$ and a covariance matrix $\mathrm{Sym}_d^+$, we indicate the correspondent Gaussian probability measure with $\mathcal{N}(m, \Sigma)$, and with $\mathcal{N}^d \subset \mathcal{P}_2(\mathbb{R}^d)$ the set of all Gaussian probability measures over $\mathbb{R}^d$. For any $A, B \in \mathrm{Sym}_d$ we write $A \succeq B$ if $A - B \in \mathrm{Sym}_d^+$ and $A \succ B$ if $A - B \in \mathrm{Sym}_d^{++}$. The trace operator is given by $A \in \mathbb{R}^{d \times d}$, $\mathrm{tr}(A) = \sum_{i=1}^d A_{ii}$, and for $A \in \mathrm{Sym}_d^{++}$, $\sqrt{A} = B$ where $B$ is the unique matrix $B \in \mathrm{Sym}_d^{++}$ such that $BB = A$.

Random variables are assumed to be defined on a common probability space $(\Omega, \mathcal{F}, \mathbb{P})$. We write $X \sim \mu$, if $X$ is a random variable with law $\mu \in \mathcal{P}(\mathbb{R}^d)$.

## A.2. Wasserstein distance

For two $\mu, \nu \in \mathcal{P}_2(\mathbb{R}^d)$ the $L^2$-Wasserstein distance (Santambrogio, 2015) is defined as

$$\mathbb{W}(\mu, \nu) := \sqrt{\inf_{X \sim \mu, Y \sim \nu} \mathbb{E}|X - Y|^2} \, .$$

Given two Gaussians $\mu = \mathcal{N}(m, \Sigma)$ and $\nu = \mathcal{N}(\overline{m}, \overline{\Sigma})$ which are non-singular, that is, $\Sigma, \overline{\Sigma} \in \mathrm{Sym}_d^{++}$, the $L^2$-Wasserstein distance takes the explicit form (Givens & Shortt, 1984; Olkin & Pukelsheim, 1982; Dowson & Landau, 1982)

$$\mathbb{W}(\mu, \nu) = \sqrt{|m - \overline{m}|^2 + \mathrm{tr}(\Sigma) + \mathrm{tr}(\overline{\Sigma}) - 2\mathrm{tr}\left(\Sigma^{1/2}\overline{\Sigma}\Sigma^{1/2}\right)^{1/2}} \, .$$

With a slight abuse of notation, for two zero-mean Gaussian measures we will sometimes use the shorter $\mathbb{W}(\Sigma, \overline{\Sigma})$ to indicate $\mathbb{W}\left(\mathcal{N}(0, \Sigma), \mathcal{N}(0, \overline{\Sigma})\right)$. In the literature, this is referred to as the Bures–Wasserstein distance, as it also coincides with the Bures metric (Uhlmann, 1992) between covariance matrices in quantum information theory.

Let $\omega : \mathcal{P}(\mathbb{R}^d) \to [0, +\infty)$ be a weight function. Given a collection of probability measures $\mu^1, \dots, \mu^N \in \mathcal{P}_2(\mathbb{R}^d)$, the weighted Wasserstein barycenters, or Frechét means, are defined as the solutions to the problem

$$\overline{\mu} \in \operatorname*{argmin}_{\nu \in \mathcal{P}_2(\mathbb{R}^d)} \sum_{i=1}^N \omega(\mu^i) \mathbb{W}^2(\mu^i, \overline{\mu}) \, . \tag{15}$$

The notion of barycenter generalizes the notion of mean to metric spaces, and uniqueness in the Wasserstein space is ensured only in presence of probability densities, see (Agueh & Carlier, 2011) for more details. Moreover, if the measures are non-singular Gaussians, $\mu^i = \mathcal{N}(m^i, \Sigma^i)$, $i = 1, \dots, N$, the barycenter is unique and it is also a Gaussian $\overline{\mu} = \mathcal{N}(\overline{m}, \overline{\Sigma})$ with mean and covariance matrix characterized by the equations

$$\overline{m} = \sum_{i=1}^N \frac{\omega(\mu^i)}{\sum_j \omega(\mu^j)} m^i, \qquad \overline{\Sigma} = \sum_{i=1}^N \frac{\omega(\mu^i)}{\sum_j \omega(\mu^j)} \left((\Sigma^i)^{1/2}\overline{\Sigma}(\Sigma^i)^{1/2}\right)^{1/2} \, . \tag{16}$$

We note that the barycenter mean takes an explicit form, while the covariance matrix is the solution of a matrix equation, which is well-defined (Rüschendorf & Uckelmann, 2002). The equation can be used for the computation of the barycenter via a fixed point iteration (Álvarez Esteban et al., 2016) which is proven to convergence with a rate (Chewi et al., 2020). This strategy corresponds to solving (15) through a (Wasserstein) gradient descent algorithm with step size 1, see (Zemel & Panaretos, 2019).

### A.3. Bures–Wasserstein manifold

The convenience of working with Gaussian measures goes beyond having explicit formulas for $\mathbb{W}$ and simpler characterization of barycenters. The space of non-singular Gaussians with the $L^2$-Wasserstein metric attains the much richer structure of a Riemannian manifold, known as the Bures-Wasserstein (BW) manifold (Takatsu, 2011; Malagò et al., 2018; Bhatia et al., 2019).

We identify the space of non-singular Gaussian measure with $\mathbb{R}^d \times \mathrm{Sym}_d^{++}$. At every point $(m, \Sigma) \in \mathbb{R}^d \times \mathrm{Sym}_d^{++}$, the tangent space is given by $T_{(m,\Sigma)}(\mathbb{R}^d \times \mathrm{Sym}_d^{++}) = \mathbb{R}^d \times T_\Sigma \mathrm{Sym}_d^{++}$ with

$$T_\Sigma \mathrm{Sym}_d^{++} = \mathrm{Sym}_d .$$

In the literature, one can find two different parametrizations of the BW Riemannian metric on $T_\Sigma \mathrm{Sym}_d^{++}$: one introduced in (Takatsu, 2011), and the other one studied in (Malagò et al., 2018; Bhatia et al., 2019). We use the former as it consistent with the classical Riemmanian-like structure of the $L^2$-Wasserstein space introduced in the seminal paper by F. Otto for the porous medium equation (Otto, 2001). Moreover, it allows for more efficient computations of exponential maps, see Remark A.2 for a comparison with the alternative parametrization presented in (Malagò et al., 2018; Bhatia et al., 2019). For any $T, S \in T_\Sigma \mathrm{Sym}_d^{++}$, the Riemannian metric is given by

$$d_\Sigma(T, S) = \mathrm{tr}(T\Sigma S) =: \langle T, S \rangle_\Sigma . \tag{17}$$

The Riemannian exponential map is defined by

$$\exp_\Sigma(T) := (I + T)\Sigma(I + T) \qquad \text{for} \quad T \succ -I . \tag{18}$$

The condition $T \succ -I$ is required to ensure that $(I + T)\Sigma(I + T) \succ 0$, that is, $\exp_\Sigma(T) \in \mathrm{Sym}_d^{++}$. Therefore, the BW manifold is not geodesically complete and the definition domain of the exponential is the translated open cone $\mathrm{Sym}_d^{++} - I$. The Riemannian logarithm between $\Sigma, \overline{\Sigma} \in \mathrm{Sym}_d^{++}$ is given by

$$\log_\Sigma(\overline{\Sigma}) := \overline{\Sigma}(\Sigma^i \overline{\Sigma})^{-\frac{1}{2}} - I , \tag{19}$$

and corresponds to the optimal transport map (shifted by $I$) between $\mathcal{N}(0, \Sigma)$ and $\mathcal{N}(0, \overline{\Sigma})$, that is,

$$\mathbb{W}^2(\Sigma, \overline{\Sigma}) = \mathbb{E}|X - (I + \log_\Sigma(\overline{\Sigma}))X|^2 \qquad \text{with} \quad X \sim \mathcal{N}(0, \Sigma) .$$

The restriction $T \succ -I$ on the exponential map $T$ becomes now intuitive in light of Brenier's theorem (Brenier, 1991): as the optimal transport map needs to be the gradient of a convex function, we have indeed that $x^\top(I + T)x$ is convex only provided $I + T \succeq 0$. The case where $I + T$ is only positive semi-definite is excluded as it would transport $\mathcal{N}(0, \Sigma)$ to a singular Gaussian, thus leaving the BW manifold. More generally, we underline that the BW geometry is coherent with the formal Otto Riemannian geometry (Otto, 2001), which is actually rigorous when restricting to the space of probabilities with smooth positive densities (Lott, 2008).

Furthermore, the space of Gaussian measures can be seen as a stratified space of manifolds, each corresponding to a different rank of the covariance matrix (Thanwerdas & Pennec, 2023). While Wasserstein distance between singular Gaussians are well-defined, the Riemannian structure is lost at the boundary of the BW manifold. We note that this stratified structure of sub-manifolds appears also in the larger $L^2$-Wasserstein space, see Remark 2.1 in (Ren & Wang, 2024).

*Remark* A.1. The BW geometry is only one of the possible choices of Riemannian geometry for $\mathrm{Sym}_d^{++}$. A popular choice, for instance, is the Affine Invariant (AI) metric (Pennec et al., 2006). In (Han et al., 2021), the authors compare the two metrics concluding that BW is more suitable for optimization in $\mathrm{Sym}_d^{++}$ thanks to its non-negative curvature and linear dependence on the space. Also, we note that the exponential maps in the form (18) is computationally cheap to evaluate. This is an important feature that allow us to parameterize the space $\mathrm{Sym}_d^{++}$ via tangent vectors in the Linearized Bures–Wasserstein geometry.

*Remark* A.2. In (Malagò et al., 2018; Bhatia et al., 2019) the authors propose a different definition of the Riemmanian metric on $T\mathrm{Sym}_d^{++}$ given by

$$\tilde{d}_\Sigma(V, U) := \frac{1}{2}\mathrm{tr}\left(\mathcal{L}_\Sigma[V]U\right) \tag{20}$$

for $V, U \in \mathrm{Sym}_d$ and $\Sigma \in \mathrm{Sym}_d^{++}$, where $\mathcal{L}_\Sigma[V]$ is known as the Lyapunov operator, and it is the unique solution to the Lyapunov equation $\mathcal{L}_\Sigma[V]\Sigma + \Sigma\mathcal{L}_\Sigma[V] = V$. The corresponding exponential map is studied in details in (Thanwerdas & Pennec, 2023) and it is given by

$$\widetilde{\exp}_\Sigma(V) = \Sigma + V + \mathcal{L}_\Sigma[V]\Sigma\mathcal{L}_\Sigma[V] \, .$$

The relation between the different metrics proposed, (17) and (20), is given by

$$T = \mathcal{L}_\Sigma[V] \, .$$

The literature for the computation of the Lyapunov operator is particularly rich and it includes methods for large-scale matrices too, see the review paper (Simoncini, 2016). As mentioned in (Han et al., 2021), the cost of exact computation is the same as matrix exponential or inversion, that is, $\mathcal{O}(d^3)$. The exponential map (18), instead, requires to perform matrix multiplications.

## B. Linearized Bures–Wasserstein space

In this section we propose a more detailed derivation of the Linearized Bures–Wasserstein (LBW) geometry used to define the Gaussian Consensus-Based Optimization algorithm.

### B.1. Linear Optimal Transport

The Linear Optimal Transport (LOT) distance is a computationally efficient metric between probabilities measures which has recently gained popularity in applications (Kolouri et al., 2016; Moosmüller & Cloninger, 2023; Sarrazin & Schmitzer, 2024; Cai et al., 2020). Consider a reference probability measure $\mu^0 \in \mathcal{P}_2^{ac}(\mathbb{R}^d)$, and $\mu^1, \mu^2 \in \mathcal{P}_2(\mathbb{R}^d)$. Let $\mathcal{T}_1, \mathcal{T}_2 : \mathbb{R}^d \to \mathbb{R}^d$ the OT maps from $\mu^0$ to $\mu^1, \mu^2$, respectively. That is, we have that $\mathbb{W}(\mu^0, \mu^1)^2 = \mathbb{E}|X - \mathcal{T}_1(X)|^2$ if $X \sim \mu^0$, and the same holds for $\mu^2$. The LOT distance with base $\mu^0$ is given by

$$\mathrm{LW}_{\mu^0}(\mu^1, \mu^2) := \mathbb{E}|\mathcal{T}_1(X) - \mathcal{T}_2(X)|^2 \qquad \text{for} \quad X \sim \mu^0 \, , \tag{21}$$

or, equivalently, it corresponds to the weighted $L^2$ norm $\|\mathcal{T}_1 - \mathcal{T}_2\|_{L^2(\mu^0)}$ between the OT maps. Derived as a simplified version of the Wasserstein metric, its main feature is that it allows to compute the mutual distances between $N$ probability measures by solving only $N$ optimal transport problems (instead of the $N^2$ required by $\mathbb{W}$).

We apply the LOT approach to the BW space to derive the LBW metric between Gaussian measures. As we expect, this corresponds to the BW Riemannian metric at a reference Gaussian measure $\mu^0$.

To show this, we first recall that if $Y \sim \mathcal{N}(0, \Sigma)$, then $\mathbb{E}|Y|^2 = \mathrm{tr}(\Sigma)$ and, for $A \in \mathrm{Sym}_d$, it holds $AY \in \mathcal{N}(0, A\Sigma A)$. Consider now $\mu^0 = \mathcal{N}(0, \Sigma^0), \Sigma^0 \in \mathrm{Sym}_d^{++}$, and, for simplicity, two other zero-mean measures $\mu^1 = \mathcal{N}(0, \Sigma^1), \mu^1 = \mathcal{N}(0, \Sigma^1), \Sigma^1, \Sigma^2 \in \mathrm{Sym}_d^{++}$. We note that the OT map between zero-mean Gaussians is a linear map, and in particular, from (18) and (19) it holds $\mathcal{T}_i(x) = (I + \log_{\Sigma^0}(\Sigma^i))x, i = 1, 2$. Direct computations show that the LOT distance corresponds to the BW metric at $T_{\Sigma^0}\mathrm{Sym}_d^{++}$:

$$\begin{aligned}
\mathrm{LW}_{\mu^0}(\mu^1, \mu^2)^2 &= \mathbb{E}|\mathcal{T}_1(X) - \mathcal{T}_2(X)|^2 \\
&= \mathbb{E}|(\mathcal{T}_1(X) - X) - (\mathcal{T}_2(X) - X)|^2 \\
&= \mathbb{E}\left|\left(\log_{\Sigma^0}(\Sigma^1) - \log_{\Sigma^0}(\Sigma^2)\right)X\right|^2 \\
&= \mathrm{tr}\left(\left(\log_{\Sigma^0}(\Sigma^1) - \log_{\Sigma^0}(\Sigma^2)\right)\Sigma^0\left(\log_{\Sigma^0}(\Sigma^1) - \log_{\Sigma^0}(\Sigma^2)\right)\right) \\
&= \|\log_{\Sigma^0}(\Sigma^1) - \log_{\Sigma^0}(\Sigma^2)\|_{\Sigma^0}^2
\end{aligned}$$

since $\mathrm{tr}(A\Sigma^0 B) = \langle A, B\rangle_{\Sigma^0}$ from (17). Of course, if we include arbitrary means $m^1, m^2 \in \mathbb{R}^d$, for $\mu^1 = \mathcal{N}(m^1, \Sigma^1)$, $\mu^2 = \mathcal{N}(m^2, \Sigma^2)$ the same computations lead to

$$\mathrm{LW}_{\mu^0}(\mu^1, \mu^2)^2 = |m^1 - m^2|^2 + \|\log_{\Sigma^0}(\Sigma^1) - \log_{\Sigma^0}(\Sigma^2)\|_{\Sigma^0}^2 \, . \tag{22}$$

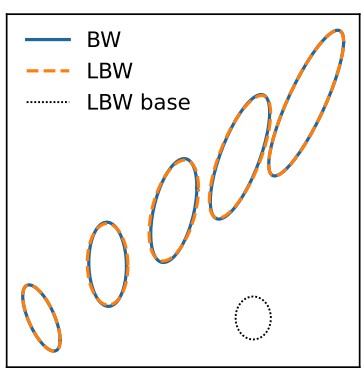
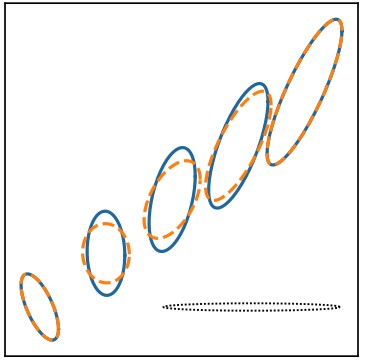
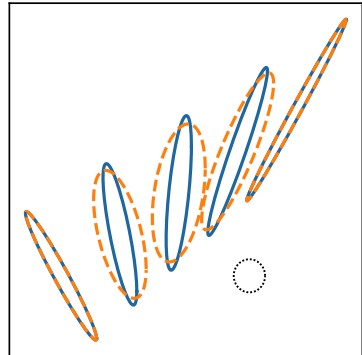

*(A)* Geodesics in the BW and LBW spaces

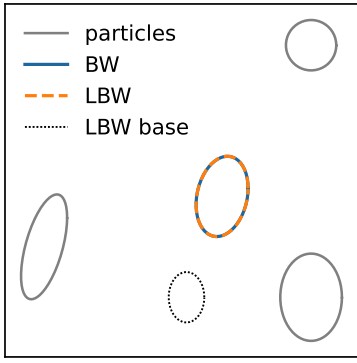
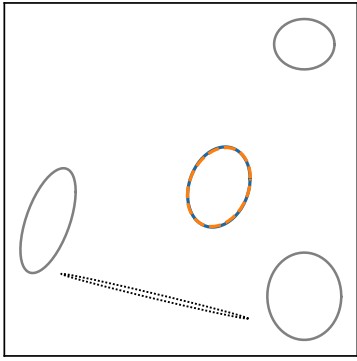
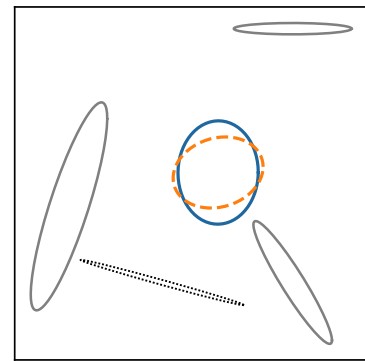

*(B)* Barycenters between 3 Gaussian particles in the BW and LBW spaces

*Figure 6.* Visual comparison between the geometry of BW and its linearization LBW. Different scenarios are considered, and, in particular, different base measures for LBW are used. Each Gaussian measure $\mathcal{N}(m, \Sigma)$ is represented by an ellipsis centered at the mean $m$ and stretched according to the covariance matrix $\Sigma$.

### B.2. Geodesics, barycenters and extension of exponential maps

Geodesics in the LBW geometry corresponds to the Wasserstein generalized geodesics (Ambrosio et al., 2008) and are given by $\mu(\tau) = \mathcal{N}(m(\tau), \Sigma(\tau))\ \tau \in [0, 1]$ with

$$m(\tau) = (1 - \tau)m^1 + \tau m^2\,, \qquad \Sigma(\tau) = \exp_{\Sigma^0}\left((1 - \tau)\log_{\Sigma^0}(\Sigma^1) + \tau \log_{\Sigma^0}(\Sigma^2)\right)\,.$$

In Figure 6A we compare BW and LBW geodesics for different values of $\Sigma^0, \Sigma^1, \Sigma^2$. The two geometries seem to diverge the more the covariance matrices are close to being singular.

Barycenters in LBW take an explicit form, unlike BW barycenters. Consider $N$ Gaussian particles $\mu^i = \mathcal{N}(m^i, \Sigma^i), i = 1, \dots, N$, their weighted LBW barycenter is directly given by $\overline{\mu} = \mathcal{N}(\overline{\mu}, \overline{\Sigma})$ with

$$\overline{m} = \sum_{i=1}^{N} \frac{\omega(\mu^i)}{\sum_j \omega(\mu^j)} m^i\,, \qquad \overline{\Sigma} = \exp_{\Sigma^0}\left(\sum_{i=1}^{N} \frac{\omega(\mu^i)}{\sum_j \omega(\mu^j)} \log_{\Sigma^0}(\Sigma^i)\right)\,. \qquad (23)$$

Note that if we identify $\Sigma^i$ with the corresponding tangent vector $T^i = \log_{\Sigma^0}(\Sigma^i)$ the characterization of the barycenter reduces to

$$\overline{T} = \sum_{i=1}^{N} \frac{\omega(\mu^i)}{\sum_j \omega(\mu^j)} T^i\,. \qquad (24)$$

While it may not be obvious by their respective expressions (16) and (23), the LBW barycenter corresponds to a first order approximation to the BW one. As shown in (Zemel & Panaretos, 2019), the LBW barycenter vector $\overline{T}$ corresponds to the

(negative) gradient of the BW barycenter functional (15). As we can see from Figure 6B, this approximation can be fairly accurate if, again, the Gaussian particles are far from the singularity.

Following the LOT approach, we are identifying each non-degenerate Gaussian measure with its corresponding BW tangent vector. Since the BW manifold is geodesically incomplete and the exponential domain is $\mathrm{Sym}_d^{++} - I$ (not the entire tangent space $\mathrm{Sym}_d$), the parametrization of the covariance matrices is given by

$$(\mathrm{Sym}_d^{++} - I) \to \mathrm{Sym}_d^{++}$$
$$T \mapsto \Sigma = \exp_{\Sigma^0}(T).$$

To be able to parametrize every Gaussian measure, possibly with a singular covariance matrix, we can extend by continuity the definition domain to the closed convex cone $\mathrm{Sym}_d^+ - I$. We will go further and extend its definition to the entire space $\mathrm{Sym}_d$ by simply setting

$$\exp_{\Sigma^0}(T) := (I + T)\Sigma^0(I + T) \qquad \text{for any} \quad T \in \mathrm{Sym}_d.$$

Note that the exponential always satisfies $\exp_{\Sigma^0}(T) \in \mathrm{Sym}_d^+$.

To sum up, given a base Gaussian measure $\mu^0 = \mathcal{N}(m^0, \Sigma^0)$, $m^0 \in \mathbb{R}^d, \Sigma \in \mathrm{Sym}_d^{++}$, the LBW space can be identified as the finite-dimensional Euclidean space

$$\mathbb{R}^d \times \mathrm{Sym}_d \quad \text{with product} \quad \langle m^1, m^2 \rangle + \langle T^1, T^2 \rangle_{\Sigma^0} \tag{25}$$

for any $(m^1, T^1), (m^2, T^2) \in \mathbb{R}^d \times \mathrm{Sym}_d$, and we denote with $\|\cdot\|_{\mathrm{LBW}(\Sigma^0)}$ the respective norm. While this leads to a redundant parametrization of $\mathcal{N}_d$, we obtain the computational benefit of dealing with an unconstrained space rather than the open cone $\mathrm{Sym}_d^{++} - I$. This is particularly convenient for the definition of Brownian Gaussian particles.

### B.3. Brownian processes in LBW

We recall for completeness how we constructed a Brownian process in LBW.

Note that the dimension of $\mathrm{Sym}_d$ is $d(d+1)/2$ and that, for $\Sigma^0 = I$ an orthogonal basis is simply given by symmetric matrices with either only one non-zero entry at the diagonal, or two non-zero entries off the diagonal. We refer to Section Appendix D for a discussion on how to generate an orthonormal basis $\{e_\ell\}_{\ell=1}^{d(d+1)/2}$ for an arbitrary $\Sigma^0 \in \mathrm{Sym}_d^{++}$. Consider now a Brownian process $(B_t^m)_{t \geq 0}$ in $\mathbb{R}^d$ and $d(d+1)/2$ i.i.d. one-dimensional Brownian processes $\{(\xi^\ell)_{t \geq 0}\}_{\ell=1}^{d(d+1)/2}$. We can construct a Brownian process in LBW as

$$B_t := (B_t^m, B_t^T) \qquad \text{where} \quad B_t^T := \sum_{\ell=1}^{d(d+1)/2} e_\ell \, \xi_t^\ell. \tag{26}$$

It is interesting to note that the associated probability measure

$$\mu_t = \mathcal{N}(B_t^m, \Sigma_t) \qquad \text{with} \quad \Sigma_t = \exp_{\Sigma^0}(B_t^T)$$

is a random process taking values in the space $\mathcal{N}_d$ of Gaussian measures. Moreover, it explores the entire $\mathcal{N}_d$, including Gaussians with singular covariance matrix, going therefore beyond the BW space. Figure 2 shows a Brownian path $B_t^T$ and the corresponding covariance matrix $\Sigma_t$ via a 2-dimensional projection of the dynamics.

*Remark* B.1. In principle, one could choose to restrict the LBW space to the closed cone of optimal transport maps, that is, constrain the tangent vector $T \in \mathrm{Sym}_d$ to the closed convex cone $\mathrm{Sym}_d^+ - I$ by including suitable boundary conditions. The random process exploring the space would then consist of a reflected SDE (Pilipenko, 2014) whose numerical simulation requires to project the dynamics back to the cone at every time iteration (Pettersson, 2000). Therefore, for computational efficiency, we chose here to simply extend the domain of the map $\exp_{\Sigma^0}(\cdot)$ to the entire space $\mathrm{Sym}_d$, and to consider unconstrained dynamics.

## C. Well-posedness and convergence: proofs and additional remarks

We recall in this section the main assumptions under which the particles CBO dynamics (7) and its mean-field approximation (11) are well-defined, and provide the proofs. We also discuss additional convergence techniques for the CBO-type algorithms (see Remark C.4).

## C.1. Particles dynamics: Lemma 3.1 and Lemma 3.2

In Lemma 3.1 we claimed that the particle evolution (7) admits a unique strong solution provided (8) holds, which we recall was

$$|\mathcal{E}(\mu^1) - \mathcal{E}(\mu^2)| \leq L_{\mathcal{E}} \left(1 + M_2(\mu^1) + M_2(\mu^2)\right) \mathrm{LW}_{\mu^0}(\mu^1, \mu^2).$$

*Proof of Lemma 3.1.* Recall from (25) that the LBW norm on $\mathbb{R}^d \times \mathrm{Sym}_d$ with base $\Sigma^0$ is given by $\|(m, T)\|^2_{\mathrm{LBW}(\Sigma^0)} = |m|^2 + \|T\|^2_{\Sigma^0}$. We first notice that the locally Lipschitz continuity assumption on $\mathcal{E}^\#$ is equivalent to show

$$|\mathcal{E}^\#(m^1, T^1) - \mathcal{E}^\#(m^2, T^2)| \leq L_{\mathcal{E}} \left(1 + \|(m^1, T^1)\|_{\mathrm{LBW}(\Sigma^0)} + \|(m^2, T^2)\|_{\mathrm{LBW}(\Sigma^0)}\right)$$
$$\times \|(m^1, T^1) - (m^2, T^2)\|_{\mathrm{LBW}(\Sigma^0)}.$$

This is also an equivalent condition to (8) since $M_2(\mu^i)$ and $\|(m^i, T^i)\|_{\mathrm{LBW}(\Sigma^0)}$ are equivalent up to a positive constant:

$$M_2(\mu^i)^2 = |m^i|^2 + \mathrm{Tr}((I + T^i)\Sigma^0(I + T^i)) = |m^i|^2 + \|I + T^i\|^2_{\Sigma^0}, \tag{27}$$

and since $\mathrm{LW}_{\mu^0}(\mu^1, \mu^2) = \|(m^1, T^1) - (m^2, T^2)\|_{\mathrm{LBW}(\Sigma^0)}$. After identifying $\mathbb{R}^d \times \mathrm{Sym}_d$ with $\mathbb{R}^D$ for given an orthonormal basis, we can use the well-posedness result, Theorem 2.1 in (Carrillo et al., 2018), which states that the dynamics is well-posed for a locally Lipschitz, finite-dimensional, objective function $\mathcal{E}^\#$. $\qquad\square$

Next, we claimed in Lemma 3.2 that a regularized version of the KL divergence, $\mathrm{KL}_\varepsilon$ satisfies the locally Lipschitz condition (8) needed for well-posedness. We recall that for a target measure $\mu^{\mathrm{targ}} \propto \exp(-V)$ the KL divergence is given for $\mu \in \mathcal{P}^{ac}(\mathbb{R}^d)$ by

$$\mathrm{KL}(\mu|\mu^{\mathrm{targ}}) = \mathcal{V}(\mu) + \mathcal{U}(\mu) = \int V(x)\mu(\mathrm{d}x) + \int \log(\mu(x))\mu(\mathrm{d}x).$$

In Section 3 we considered a regularized version $\mathcal{U}_\varepsilon, \varepsilon > 0$, for the log-entropy $\mathcal{U}$ defined as

$$\mathcal{U}_\varepsilon(\mu) = \mathcal{U}\left(\mathcal{N}(m, \mathrm{clip}_\varepsilon(\Sigma))\right) \qquad \text{for} \quad \mu = \mathcal{N}(\mu, \Sigma),$$

where $\mathrm{clip}_\varepsilon(\Sigma) := \sum_\ell \max\{\lambda_\ell, \wedge \varepsilon\} u_\ell u_\ell^\top$ for $\Sigma = \sum_\ell \lambda_\ell u_\ell u_\ell^\top$, with $(\lambda_\ell, u_\ell)_\ell$ an eigenbasis for $\Sigma$. The associated regularized KL divergence for Gaussians $\mu$ is then given by

$$\mathrm{KL}_\varepsilon(\mu|\mu^{\mathrm{targ}}) = \mathcal{V}(\mu) + \mathcal{U}_\varepsilon(\mu).$$

We recall first a result from (Polyanskiy & Wu, 2016)

**Proposition C.1** ((Polyanskiy & Wu, 2016), Proposition 1). *Let $\mu^1, \mu^2 \in \mathcal{P}_2^{ac}(\mathbb{R}^d)$, and $(c_1, c_2)$-regular, that is, such that*

$$|\nabla \log \mu^i(x)| \leq c_1|x| + c_2,$$

*then*

$$|\mathcal{U}(\mu^1) - \mathcal{U}(\mu^1)| \leq \left(c_2 + \frac{c_1}{2}M_2(\mu^1) + \frac{c_1}{2}M_2(\mu^2)\right) \mathbb{W}(\mu^1, \mu^2). \tag{28}$$

*Proof of Lemma 3.2.* We need to check that $\mathrm{KL}_\varepsilon$ satisfied the locally Lipschitz condition (8). First of all, we note that since the LOT distance is an upper bound for the $L^2$-Wasserstein distance, condition

$$|\mathcal{E}(\mu^1) - \mathcal{E}(\mu^2)| \leq L_{\mathcal{E}} \left(1 + M_2(\mu^1) + M_2(\mu^2)\right) \mathbb{W}(\mu^1, \mu^2) \tag{29}$$

implies (8). We show that both $\mathcal{V}$ and $\mathcal{U}_\varepsilon$ satisfy (29) and, as a consequence, so does $\mathrm{KL}_\varepsilon$.

Note that $\mathcal{V}(\mu) = \mathbb{E}V(X)$ for $X \sim \mu$, and let $\mu^1, \mu^2 \in \mathcal{N}^d$ and $X^1 \sim \mu^1, X^2 \sim \mu^2$ such that are they are optimally coupled: $\mathbb{E}|X^1 - X^2|^2 = \mathbb{W}^2(\mu^2)$. It holds

$$
\begin{aligned}
|\mathcal{V}(\mu^1) - \mathcal{V}(\mu^2)| &= |\mathbb{E}V(X^1) - \mathbb{E}V(X^2)| \\
&\leq L_V \mathbb{E}\left[(1 + |X^1| + |X^2|)|X^1 - X^2|\right] \\
&\leq L_V \sqrt{\mathbb{E}\left[(1 + |X^1| + |X^2|)^2\right]}\sqrt{\mathbb{E}|X^1 - X^2|^2} \\
&\leq CL_V \left((1 + \sqrt{\mathbb{E}|X^1|^2} + \sqrt{\mathbb{E}|X^1|^2})\right)\mathbb{W}(\mu^1, \mu^2)
\end{aligned}
$$

for some $C > 0$, where we used Cauchy–Schwartz inequality and the coupling optimality. Since $\sqrt{\mathbb{E}|X^i|^2} = M_2(\mu^i)$, we proved that $\mathcal{V}$ is locally Lipschitz in the sense of (29).

For $\mu = \mathcal{N}(m, \Sigma)$ with $\Sigma \succcurlyeq \varepsilon I$, $\varepsilon > 0$, it holds

$$|\nabla \log \mu(x)| = |\Sigma^{-1}(x - m)| \leq \varepsilon^{-1}(|x| + |m|),$$

sine the largest eigenvalue of $\Sigma^{-1}$ is bounded by $\varepsilon^{-1}$. By applying Proposition C.1 we then obtain that (29) is satisfied uniformly for Gaussians with bounded eigenvalues. $\qquad\square$

We conclude with a remark on other possible energy functionals.

*Remark* C.2. An alternative discrepancy measure to the KL divergence is the Maximum Mean Discrepancy (MMD) induced by a reproducing kernel Hilbert space (RKHS). Under suitable smoothness and normalization conditions on the kernel, one can prove that the MMD is controlled by the Wasserstein distance. In particular, as shown in (Vayer & Gribonval, 2023), Proposition 2, Corollary 3, it holds

$$\|\mu^1 - \mu^2\|_{\mathcal{H}_\kappa} \leq C\mathbb{W}(\mu^1, \mu^2),$$

for a constant $C$ depending on the curvature of the kernel at the origin. We refer to (Vayer & Gribonval, 2023) and the references therein for more details and the definition of the norm $\|\cdot\|_{\mathcal{H}_\kappa}$ associated to the reproducing kernel space $\mathcal{H}_\kappa$.

### C.2. Proof of Lemma 3.4

Recall in Lemma 3.4 we stated that the mean-field dynamics (11) admits a unique strong solution provided $\mathcal{E}$ satisfies Assumption 3.3. That is, if $\mathcal{E}$ is bounded form below, and either bounded from above or grows quadratically at infinity

$$\mathcal{E}(\mu) - \inf \mathcal{E} \geq c_l M_2(\mu)^2 \qquad \text{for} \quad \mu, \quad M_2(\mu) > R. \tag{30}$$

*Proof of Lemma 3.4.* As in Lemma 3.1, well-posedness of the mean-field dynamics follows from well-posedness of the mean-field CBO dynamics in $\mathbb{R}^D$. In (Carrillo et al., 2021) the authors prove well-posedness provided the finite-dimensional objective $\mathcal{E}^\#$ satisfies

i) $\underline{\mathcal{E}} = \inf \mathcal{E}^\# > -\infty$;

ii) there exists constants $\tilde{L}_\mathcal{E}, \tilde{c}_u$ such that for all $z_1 = (m^1, T^1), z_2 = (m^2, T^2)$

$$\begin{cases} |\mathcal{E}^\#(z_1) - \mathcal{E}^\#(z_2)| \leq \tilde{L}_\mathcal{E} \left(1 + \|z_1\|_{\text{LBW}(\Sigma^0)} + \|z_2\|_{\text{LBW}(\Sigma^0)}\right) \|z_1 - z_2\|_{\text{LBW}(\Sigma^0)} \\ \mathcal{E}^\#(z_1) - \underline{\mathcal{E}} \leq \tilde{c}_u \left(1 + \|z_1\|_{\text{LBW}(\Sigma^0)}^2\right), \end{cases}$$

iii) either $\sup \mathcal{E}^\# < +\infty$ or there exists $\tilde{M}, \tilde{c}_l$ such that for all $z$

$$\mathcal{E}^\#(z) - \underline{\mathcal{E}} \geq \tilde{c}_l \|z\|_{\text{LBW}(\Sigma^0)}^2 \qquad \text{for} \quad \|z\|_{\text{LBW}(\Sigma^0)} \geq \tilde{R},$$

see Assumption 3.1, Theorem 3.1, and Theorem 3.2 in (Carrillo et al., 2021). Condition i) is equivalent to what we have assumed in Assumption 3.3. We note that condition ii) is a small modification of (Carrillo et al., 2021), Assumption 3.1, where the Lipschitz constant takes the form $(\|z_1\| + \|z_2\|)$, but this change does have an impact on the proof. Also, the quadratic upper bound follows from the local Lipschitz assumption. Altogether, thanks to the equivalence between the LBW norm and the second moment $M_2(\mu)$ for the corresponding measure $\mu$ (see (27)), then condition ii) follows form the locally Lipschitz condition (8) on $\mathcal{E}$. For the same reason, also the quadratic lower bound iii) for $\mathcal{E}^\#$ is equivalent (up to constants) to the quadratic lower bound (30) in Assumption 3.3 for $\mathcal{E}$. $\qquad\square$

We now check more precisely under which condition $\mathcal{V}, \mathcal{W}$ grow quadratically at infinity as condition (30), which appears in Assumption 3.3. Note that, since the regularized log-entropy $\mathcal{U}_\varepsilon$ is bounded, quadratic growth of $\mathcal{V}$ also implies quadratic growth of $\text{KL}_\varepsilon(\cdot|\mu^{\text{targ}})$ for $\mu^{\text{targ}} \propto \exp(-V)$.

**Lemma C.3.** *The growth condition* (30) *is satisfied for the energies* $\mathcal{V}, \mathcal{W}$ *provided*

$$V(x) - \inf V \geq 2c_l|x|^2 \qquad\qquad \text{for} \quad |x| > R,$$
$$W(x,y) - \inf W \geq 2c_l'(|x|^2 + |y|^2) \qquad\qquad \text{for} \quad |x|^2 + |y|^2 > R,$$

*for some constants* $c_l, c_l', R > 0$ *and* $\inf V, \inf W > \infty$.

*Proof.* Let $\mu$ such that $M_2(\mu) \geq R/\sqrt{2}$, it holds

$$
\begin{aligned}
\int V(x)\mu(\mathrm{d}x) - \inf V &\geq 2c_l \int_{|x|>R} |x|^2 \mu(\mathrm{d}x) + \int_{|x|\leq R} (V(x) - \inf V)\mu(\mathrm{d}x) \\
&\geq 2c_l \int_{|x|>R} |x|^2 \mu(\mathrm{d}x) \pm 2c_l \int_{|x|\leq R} |x|^2 \mu(\mathrm{d}x) \\
&\geq 2c_l \int |x|^2 \mu(\mathrm{d}x) - 2c_l R^2 \\
&\geq 2c_l M_2(\mu)^2 - c_l M_2(\mu)^2 = c_l M_2(\mu)^2
\end{aligned}
$$

where in the second line we used that $V(x) - \inf V \geq 0$.

For $\mathcal{W}$, note that, differently from $\mathcal{V}$, in general $\inf W \neq \inf \mathcal{W}$, and $\inf W \leq \inf \mathcal{W}$. Consider

$$M_2(\mu) \geq \max\{R/\sqrt{2}, \Delta\mathcal{W}/c_l'\}, \qquad \text{where} \quad \Delta\mathcal{W} := \inf \mathcal{W} - \inf W \geq 0.$$

Similar computations as above lead to the following

$$
\begin{aligned}
\int W(x,y)&\mu(\mathrm{d}x)\mu(\mathrm{d}y) - \inf \mathcal{W} = \int W(x,y)\mu(\mathrm{d}x)\mu(\mathrm{d}y) - \inf W - \Delta\mathcal{W} \\
&\geq 2c_l' \iint_{|x|^2+|y|^2>R} (|x|^2 + |y|^2)\mu(\mathrm{d}x)\mu(\mathrm{d}y) \\
&\quad + 2c_l' \iint_{|x|^2+|y|^2\leq R} (W(x,y) - \inf W)\mu(\mathrm{d}x)\mu(\mathrm{d}y) - \Delta\mathcal{W} \\
&\geq 2c_l' \iint_{|x|^2+|y|^2>R} (|x|^2 + |y|^2)\mu(\mathrm{d}x)\mu(\mathrm{d}y) \pm 2c_l \iint_{|x|^2+|y|^2\leq R} (|x|^2 + |y|^2)\mu(\mathrm{d}x)\mu(\mathrm{d}y) - \Delta\mathcal{W} \\
&\geq 4c_l' M_2(\mu)^2 - 4c_l R^2 - \Delta\mathcal{W} \\
&\geq c_l'(4 - 2 - 1)M_2(\mu)^2 \\
&= c_l' M_2(\mu)^2.
\end{aligned}
$$

$\square$

### C.3. Additional remarks on assumptions to Theorem 3.5

In Theorem 3.5, we assume the energy function $\mathcal{E}$ we aim to minimize is differentiable with respect to the Gaussian parameters. We recall in the following when this holds for $\mathcal{V}, \mathcal{W}, \mathcal{U}$. In Remark C.4 we also discuss an alternative convergence proof for CBO-type dynamics proposed in (Fornasier et al., 2024).

**Differentiability of $\mathcal{V}$.** Let $\mu = \mathcal{N}(m, \Sigma)$, we have (see, for instance, Appendix D in (Khan & Rue, 2023))

$$\nabla_m \mathcal{V}(\mu) = \int \nabla V(x)\,\mu(\mathrm{d}x) \quad \text{and} \quad \nabla_m^2 \mathcal{V}(\mu) = \int \nabla^2 V(x)\,\mu(\mathrm{d}x).$$

With respect to the covariance,

$$\partial_{\Sigma_{ij}} \mathcal{V}(\mu) = c_{ij} \int \partial_{ij}^2 V(x)\,\mu(\mathrm{d}x) \qquad \text{with} \quad c_{ij} = \begin{cases} 1/2 & \text{if } i \neq j \\ 1 & \text{otherwise} \end{cases}$$

and consequently $\partial_{\Sigma_{ij}\Sigma_{\ell k}}^2 \mathcal{V}(\mu) = c_{ij}c_{\ell k} \int \partial_{ij\ell k}^4 V(x)\,\mu(\mathrm{d}x)$. Hence, if $V \in \mathcal{C}^4(\mathbb{R}^d)$ with bounded second- and fourth-order derivatives, then $\mathcal{V}$ satisfies the regularity assumptions in Theorem 3.5.

**Differentiability of $\mathcal{W}$.** To compute the derivatives of $\mathcal{W}$, we view $\mu \otimes \mu$ as a Gaussian measure on $\mathbb{R}^d \times \mathbb{R}^d$ with mean $(m, m)$ and block–diagonal covariance $\mathrm{diag}(\Sigma, \Sigma)$, so we can re-use the computations done for $\mathcal{V}$ variable-wise. Recall $\mathcal{W}(\mu)$ is defined in (9), so we have

$$\nabla_m \mathcal{W}(\mu) = \iint \left( \nabla_x W(x, y) + \nabla_y W(x, y) \right) \mu(\mathrm{d}x)\mu(\mathrm{d}y),$$

$$\nabla_m^2 \mathcal{W}(\mu) = \iint \left( \nabla_{xx}^2 W(x, y) + \nabla_{yy}^2 W(x, y) + \nabla_{xy}^2 W(x, y) + \nabla_{yx}^2 W(x, y) \right) \mu(\mathrm{d}x)\mu(\mathrm{d}y).$$

With respect to the covariance, we compute

$$\partial_{\Sigma_{ij}} \mathcal{W}(\mu) = c_{ij} \iint \left( \partial_{x_i x_j}^2 W(x, y) + \partial_{y_i y_j}^2 W(x, y) \right) \mu(\mathrm{d}x)\mu(\mathrm{d}y),$$

with $c_{ij}$ as before, and consequently

$$\partial_{\Sigma_{ij}\Sigma_{\ell k}}^2 \mathcal{W}(\mu) = c_{ij}c_{\ell k} \iint \left( \partial_{x_i x_j x_\ell x_k}^4 W(x, y) + \partial_{y_i y_j y_\ell y_k}^4 W(x, y) \right.$$
$$\left. + \partial_{x_i x_j}^2 \partial_{y_\ell y_k}^2 W(x, y) + \partial_{y_i y_j}^2 \partial_{x_\ell x_k}^2 W(x, y) \right) \mu(\mathrm{d}x)\mu(\mathrm{d}y).$$

**Differentiability of $\mathcal{U}$.** We recall from completeness also the case of the log-entropy which we already discusses in Section 3.

The log-entropy functional $\mathcal{U}$ is invariant under translations of the mean, which implies that its gradient with respect to $m$ vanishes, i.e. $\nabla_m \mathcal{U}(\mu) = 0$. For covariance matrices $\Sigma \in \mathrm{Sym}_d^{++}$, it is shown in Appendix A.1 of (Lambert et al., 2022) that

$$\nabla_\Sigma \mathcal{U}(\mu) = -\tfrac{1}{2} \Sigma^{-1}.$$

Furthermore, letting $e_\ell$ denote the $\ell$-th canonical basis vector of $\mathbb{R}^d$, the derivative of the matrix inverse is given by (see (Giles, 2008))

$$\frac{\partial}{\partial \Sigma_{ij}} \Sigma^{-1} = -\Sigma^{-1} e_i e_j^\top \Sigma^{-1}.$$

As a result, the boundedness assumptions on second-order derivatives required in Theorem 3.5 are only valid as long as the covariance matrix remains uniformly positive definite, that is, within a region where $\Sigma \succ \varepsilon I$ for some $\varepsilon > 0$.

This condition can be enforced in practice by applying a smooth eigenvalue-clipping procedure to $\Sigma$, as described in (10), which keeps the covariance in the admissible set and thereby guarantees convergence to global minimizers.

*Remark* C.4. For CBO methods, a different type of convergence result was proposed (Fornasier et al., 2024). We discuss briefly the assumption considered there, and why their result is not directly applicable in our settings. The finite-dimensional objective function $\mathcal{E}^\#$ is not required to be differentiable but it requires to attain a unique global minimum $z^\star$ and to attain an inverse continuity assumption around $z^\star$, see Definition 3.5 in (Fornasier et al., 2024). In particular, there must exists $\mathcal{E}_\infty, R_0, \eta > 0, \nu \in (0, \infty)$ such that

$$\|z - z^\star\|_{\mathrm{BW}(\Sigma^0)} \leq \frac{1}{\eta} \left( \mathcal{E}^\#(z) - \mathcal{E}^\#(z^\star) \right)^\nu$$

if $\|z\|_{\mathrm{BW}(\Sigma^0)} \leq R_0$ and $\mathcal{E}^\#(z) > \mathcal{E}^\#(z^\star) + \mathcal{E}_\infty$ otherwise. In terms of the functional $\mathcal{E}$, this translates into the condition

$$\mathrm{L}\mathbb{W}_{\mu^0}(\mu, \mu^\star) \leq \frac{1}{\eta} \left( \mathcal{E}(\mu) - \mathcal{E}(\mu^\star) \right)^\nu,$$

which is difficult to check. It is interesting to note that, for $\mathcal{E} = \mathrm{KL}(\cdot \,|\, \mu^{\mathrm{targ}})$ and if the solution coincides both with the target and the reference measure, $\mu^\star = \mu^{\mathrm{targ}} = \mu^0$, then the above condition is implied by the Talagrand's inequality (Otto & Villani, 2000) with constant $\lambda > 0$:

$$\mathbb{W}(\mu, \mu^{\mathrm{targ}}) \leq \sqrt{\frac{2}{\lambda} \left( \mathrm{KL}(\mu \,|\, \mu^{\mathrm{targ}}) - \mathrm{KL}(\mu^{\mathrm{targ}} \,|\, \mu^{\mathrm{targ}}) \right)}.$$

since $\mathrm{KL}(\mu^{\mathrm{targ}} \,|\, \mu^{\mathrm{targ}}) = 0$.

A natural question is whether Talagrand's inequality implies the growth condition in the LOT geometry in the case $\mu^\star \neq \mu^{\mathrm{targ}} \neq \mu^0$, but we leave it for future work. We note that lower bounds of $\mathbb{W}$ in terms of $\mathrm{L}\mathbb{W}$ have been studied in (Delalande & Merigot, 2023; Carlier et al., 2024; Letrouit & Mérigot, 2025), with (Letrouit & Mérigot, 2025), in particular, covering the case the initial measure is log-concave, and so possibly Gaussian.

## D. Numerical experiments and implementational aspects

We provide in this section more details regarding the implementational aspects of Algorithm 1, and the experiments illustrated in Section 4.

### D.1. Construction of random tangent vectors in LBW

For the Euler–Maruyama update (13) discretization of the particles dynamics, it is required to sample random normal vectors in the LBW space with base $\Sigma^0$.

Let $\Sigma^0 = I$ and $\{e_i\}_{i=1}^d$ be the canonical base of $\mathbb{R}^d$. As mentioned in Appendix B.3, an orthonormal basis with respect to $\langle \cdot, \cdot \rangle_I$ is given by the matrices

$$
J_{ij} = \begin{cases} e_i e_i^\top & \text{if } i = j, \\ (e_i e_j^\top + e_j e_i^\top)/\sqrt{2} & \text{if } i < j. \end{cases}
$$

A standard normal vector $B \in \mathrm{Sym}$ according to this basis can be conveniently generated as

$$
B = \frac{\Xi + \Xi^\top}{2} \qquad \text{with} \quad \Xi_{ij} \sim \mathcal{N}(0, 1).
$$

An orthonormal basis for $\Sigma^0 = I$ can be translated into an orthonormal basis for an arbitrary $\Sigma^0 \in \mathrm{Sym}_d^{++}$. Let $\Sigma^0 = Q\Lambda Q^\top$ be its singular value decomposition with $\Lambda = \mathrm{diag}(\lambda_1, \ldots, \lambda_d)$. An orthonormal basis for $\langle \cdot, \cdot \rangle_{\Sigma^0}$ is given by

$$
\tilde{J}_{ij} = \begin{cases} \dfrac{1}{\sqrt{\lambda_i}} Q J_{ii} Q^\top & \text{if } i = j, \\ \dfrac{1}{\sqrt{\lambda_i + \lambda_j}} Q J_{ij} Q^\top & \text{if } i < j. \end{cases}
$$

This can be checked by directly computing $\langle \tilde{J}_{ij}, \tilde{J}_{k\ell} \rangle_{\Sigma^0} = \mathrm{tr}(\tilde{J}_{ij} \Sigma^0 \tilde{J}_{k\ell})$. Analogously, a standard normal vector $\tilde{B}$ according to this basis can be constructed from a standard normal vector $B$ for the basis $\{J_{ij}\}_{i \leq j}$ by rescaling each entry:

$$
\tilde{B}_{ij} = \begin{cases} \dfrac{1}{\sqrt{\lambda_i}} B_{ii} & \text{if } i = j, \\ \dfrac{1}{\sqrt{\lambda_i + \lambda_j}} B_{ij} & \text{if } i < j, \end{cases} \qquad \text{and set } \tilde{B}_{ji} = \tilde{B}_{ij}.
$$

### D.2. Tests in $d = 2$

Figure 7 shows the solutions computed by CBO and the baseline Bures–Wasserstein Gradient Flow (GF) for a test instance where the target measure is bimodal (and hence not log-concave). The trajectory of the mean of the consensus point is plotted, as well as the final computed Gaussian, represented by an ellipse. Unlike gradient flows, the CBO trajectory follows a stochastic path, with the consensus point exploring the search space before converging to its final location. This leads to a slower initial decay of the KL divergence, but CBO ultimately finds a better solution than GF, which becomes trapped in a sub-optimal mode.

Figure 4 in Section 4 illustrated the results when testing the gradient-based baselines (see Appendix D.4) and CBO against 4 different target measure, A–D. We provide in Table 1 the detailed models parameters.

**Sensitivity analysis.** In CBO algorithms, the diffusion parameter $\sigma$ is crucial for balancing particle exploration of the search space with the emergence of consensus. Small values of $\sigma$ may lead to premature convergence, while large values

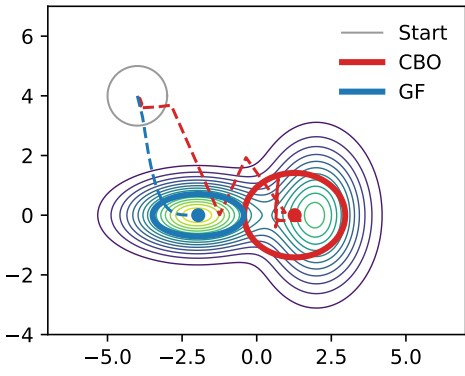 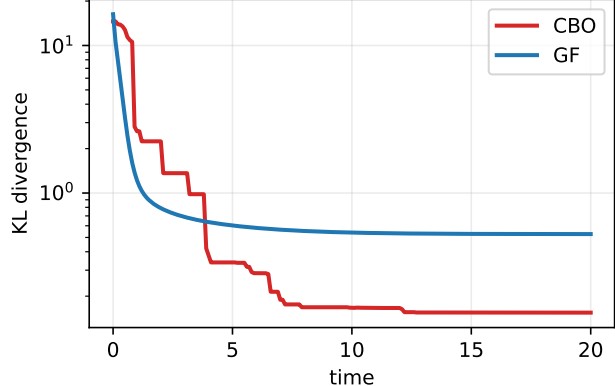

*Figure 7.* Comparison between one run of the CBO and BW GF algorithms in approximating a bimodal target density (contour lines). For CBO, the final Gaussian consensus point is shown. Trajectories of the mean of the consensus point and of the GF dynamics are also included. On the right, the evolution of the objective $\mathcal{E} = \mathrm{KL}$ shows that CBO finds a better solution to the VI problem. Parameters: $\Delta t = 0.1, \lambda = 1, \sigma = 5, N = 20, \alpha = 10^4$. Reference measure for LBW geometry: $\mathcal{N}(0, I)$.

*Table 1.* Gaussian mixture model parameters (means, covariances, and mixture weights) for targets A–D, see (14).

| Target | $K$ | means | covariances | weights |
|---|---|---|---|---|
| A | 2 | $m_1 = (-2.2, 0.0)$ 
 $m_2 = (2.2, 0.0)$ | $\Sigma_1 = \begin{pmatrix} 1 & 0.2 \\ 0.2 & 0.6 \end{pmatrix} \quad \Sigma_2 = \begin{pmatrix} 1 & -0.2 \\ -0.2 & 0.6 \end{pmatrix}$ | $w_1 = 0.5$ 
 $w_2 = 0.5$ |
| B | 2 | $m_1 = (-1.77, 1.06)$ 
 $m_2 = (-0.35, -0.35)$ | $\Sigma_1 = \begin{pmatrix} 1.25 & -0.25 \\ -0.25 & 1.25 \end{pmatrix} \quad \Sigma_2 = \begin{pmatrix} 2.50 & -1.50 \\ -1.50 & 2.50 \end{pmatrix}$ | $w_1 = 0.5$ 
 $w_2 = 0.5$ |
| C | 4 | $m_1 = (-2.47, 1.06)$ 
 $m_2 = (-1.48, 0.64)$ 
 $m_3 = (-2.05, 0.07)$ 
 $m_4 = (0.20, -1.61)$ | $\Sigma_1 = \begin{pmatrix} 0.45 & 0 \\ 0 & 0.45 \end{pmatrix} \quad \Sigma_2 = \begin{pmatrix} 1.9 & -1.9 \\ -1.9 & 2.3 \end{pmatrix}$ 
 $\Sigma_3 = \begin{pmatrix} 2.3 & -1.9 \\ -1.9 & 1.9 \end{pmatrix} \quad \Sigma_4 = \begin{pmatrix} 2.51 & -2.49 \\ -2.49 & 2.51 \end{pmatrix}$ | $w_1 = 0.25$ 
 $w_2 = 0.30$ 
 $w_3 = 0.30$ 
 $w_4 = 0.15$ |
| D | 4 | $m_1 = (-1.5, -2.0)$ 
 $m_2 = (1.5, 0.7)$ 
 $m_3 = (-1.5, 0.7)$ 
 $m_4 = (1.5, -2.0)$ | $\Sigma_1 = \Sigma_2 = \Sigma_3 = \Sigma_4 = \begin{pmatrix} 0.7 & 0 \\ 0 & 0.5 \end{pmatrix}$ | $w_1 = 0.2$ 
 $w_2 = 0.2$ 
 $w_3 = 0.2$ 
 $w_4 = 0.4$ |

can prevent convergence altogether. Values $\sigma \in [3, 5]$ yield the best performance across all test problems considered (see Figure 8A).

The number of particles $N$ is also an important choice, as it determines the trade-off between computational accuracy and efficiency. In the tests considered, however, there is little improvement beyond $N = 16$ particles, as shown in Figure 8B.

We also test the impact on the linearization procedure. So far we have kept the base measure for the LBW geometry to be the standard normal distribution $\mathcal{N}(0, I)$, that is, $\Sigma^0 = I$. To mitigate the (eventual) loss in geometry information, it is natural to think of updating the reference measure during the computation, from time to time, to linearize the space around the current consensus point. We tested this strategy for different update frequencies $\Delta t_{\mathrm{up}} > 0$, from the smallest one possible, $\Delta t_{\mathrm{up}} = \Delta t = 0.1$ to no update at all $\Delta t_{\mathrm{up}} = 12.8 > T_{\max} = 10$. As we can noticed from the results of the experiments, see Figure 8C, in the problem considered there is no benefit in updating the base measure. For the target measure C, the update strategy actually deteriorates the algorithm's accuracy. We conjecture this may be due to the numerical error introduced by frequently computing the logarithmic and exponential BW maps.

**D.3. Tests in $d = 10$**

We provide here more details regarding the experiments in $d = 10$ described in Section 4 and Figure 5.

Each target distribution is a randomly generated $K$-component Gaussian mixture model: mixture weights are sampled from

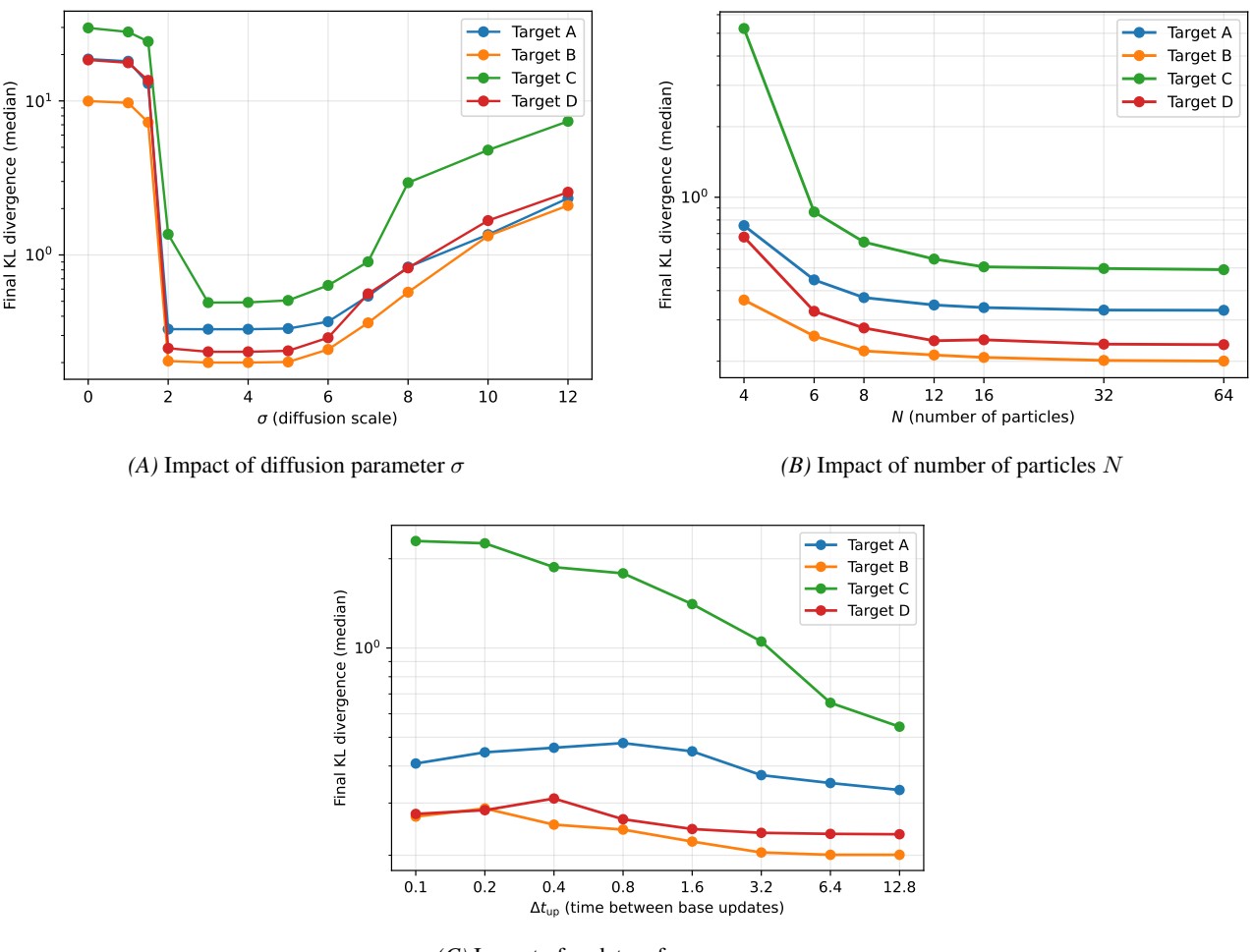

*(A)* Impact of diffusion parameter $\sigma$

*(B)* Impact of number of particles $N$

*(C)* Impact of update reference measures

*Figure 8.* Sensitivity analysis of CBO performance with respect to (A) the diffusion parameter $\sigma$, (B) the number of particles $N$, (C) update frequency of the reference measure $\mu_0 = \mathcal{N}(0, \Sigma^0)$. The curves show the median KL divergence averaged across 100 runs with different random initializations. Same parameters as experiment in Figure 4.

a Dirichlet distribution, component means are sampled uniformly on the sphere of radius $R_{\text{mean}}\sqrt{d}$ (so that components are well separated in high dimension, with $R_{\text{mean}}$ controlling the spread), and component covariances are random SPD matrices with eigenvalues in $[\lambda_{\min}, \lambda_{\max}]$.

We set $d = 10$, $K = 5$, $R_{\text{mean}} = 3.0$, $\lambda_{\min} = 0.4$, and $\lambda_{\max} = 2.0$. Both methods are initialized from a Gaussian centered at a random point near the origin with covariance equal to the identity, and we run them for a horizon $T = 75$ with step size $\Delta t = 0.1$. For CBO we employ $N = 100$ particles with parameters $\alpha = 10^4$, $\sigma = 2.5$, $\lambda = 1.0$, and we do not update the base.

To compare CBO with the different baseline methods robustly, we generate $M = 20$ independent random GMM instances and record the evolution of $\text{KL}(\mu \,|\, \mu^{\text{targ}})$ for each method. Since absolute KL values vary between instances, each trajectory is normalized by the best KL value achieved on that instance,

$$\text{RelKL}_i(t) \;=\; \frac{\text{KL}_i(t)}{K_i^\star}, \quad K_i^\star = \min_{t,m} \text{KL}_i^m(t),$$

where $i$ indexes the instance and $m \in \{\text{CBO}, \text{BW}, \text{SVGD}, \text{FR}\}$ the method. We then report the relative KL, $\text{RelKL}_i(t)$, aggregated across instances by plotting the median together with the interquartile range $[0.25, 0.75]$.

**Results.** From Figure 5, we notice that the interquantile range of CBO is smaller than that of BW and FR. We conjecture that BW and FR may sometimes get stuck in local minima, while CBO computes more robust solutions across different instances of the problem thanks to the particle exploration. On average, though, the Gaussian SVGD algorithm computes better solutions than CBO and other baselines for this class of problems.

*Remark* D.1. We note that extensions of particle-based optimizers to very high dimensions typically require additional heuristics to keep the computational cost manageable. For instance, in (Carrillo et al., 2021) a random batch technique for the computation of the consensus point was proposed to reduce the number of function evaluations per step. Another delicate aspect is the choice of the diffusion parameter $\sigma$. As noted in Section 4.3 of (Borghi et al., 2023b), the interval of values of $\sigma$ leading to good performance tends to shrink as $d$ increases, and particles become more prone either to converge prematurely or to diverge. To tackle high-dimensional machine learning problems, in (Carrillo et al., 2021) the authors also propose a heuristic in which a relatively small $\sigma$ is used, but particles are re-initialized with white noise at the end of each training epoch. Such strategies may also be applied in our context to address high-dimensional Gaussian VI problems.

### D.4. Details on baseline methods

In the experiments, we compared the proposed CBO methods with different algorithms for optimization over the space of Gaussian measures. They are single-trajectory algorithms which do not employ a set of Gaussian particles, but a single one. We considered the Bures–Wasserstein (BW) gradient flow (Lambert et al., 2022), the Gaussian Stein Variational Gradient Descent (SVGD) (Liu et al., 2023), and the natural, or Fisher–Rao (FR) gradient flow (Barfoot, 2020; Liero et al., 2025a). They all aim to minimize $\mathrm{KL}(\mu \mid \mu^{\mathrm{targ}})$ and are discretized via an explicit Euler scheme and same quadrature approximation for expected values as for CBO.

For completeness, we recall here the corresponding ODEs for the mean and covariance matrix $(m_t, \Sigma_t) \in \mathbb{R}^d \times \mathrm{Sym}_d^+$. In the following, $X_t$ is a random variable with law $\mathcal{N}(m_t, \Sigma_t)$. The objective energy to be minimized is the Kullback–Leibler divergence $\mathrm{KL}(\cdot \mid \mu^{\mathrm{targ}})$, where $\mu^{\mathrm{targ}} \propto e^{-V}$ for some potential $V \in \mathcal{C}^2(\mathbb{R}^d)$.

- The Bures–Wasserstein Gradient Flow (BW/GF) has been studied in (Lambert et al., 2022), and reads

$$\begin{cases} \dot{m}_t &= -\mathbb{E}\nabla V(X_t) \\ \dot{\Sigma}_t &= 2I_d - \mathbb{E}[\nabla V(X_t) \otimes (X_t - m_t) + (X_t - m_t) \otimes \nabla V(X_t)] \, . \end{cases} \tag{31}$$

- Gaussian Stein Variational Gradient Descent (SVGD) with kernel $K_1(x, y) = x^\top y + 1$ induces the Gaussian evolution (Liu et al., 2023)

$$\begin{cases} \dot{m}_t &= \left(I_d - \mathbb{E}\nabla^2 V(X_t)\Sigma_t\right) m_t - \left(1 + |m_t|^2\right) \mathbb{E}\nabla V(X_t) \\ \dot{\Sigma}_t &= G_t \Sigma_t + \Sigma_t G_t^\top, \qquad \text{where} \quad G_t := I_d - \mathbb{E}\nabla^2 V(X_t)\Sigma_t \, . \end{cases} \tag{32}$$

- Natural, or Fisher–Rao (FR), gradient flow (Barfoot, 2020; Liero et al., 2025a) is given by

$$\begin{cases} \dot{m}_t &= -\Sigma_t \, \mathbb{E}\nabla V(X_t), \\ \dot{\Sigma}_t &= G_t \Sigma_t + \Sigma_t G_t^\top, \qquad \text{where} \qquad G_t := \frac{1}{2}\left(I_d - \Sigma_t \, \mathbb{E}\nabla^2 V(X_t)\right) \, . \end{cases} \tag{33}$$

## E. Extension to GMM via multi-swarm approach

One may wonder whether the algorithm can be extended to optimize over the richer class of Gaussian Mixture Models (GMMs) as done in (Lambert et al., 2022) where many Gaussian particles are used to approximate the BW gradient flow. To do so, we propose a multi-swarm dynamics, inspired by particle systems for multi-objective optimization (Klamroth et al., 2024; Borghi et al., 2023a).

**Swarms' dynamics.** The full particle system is divided in $n_S$ sub-swarms, and a Gaussian CBO dynamics, analogous to single-swarm algorithm, is prescribed within the sub-swarm. The only difference lays on the definition of the objective function, which is different for every swarm. Let $\overline{\mu}^{\alpha,\ell}, \ell = 1, \dots, n_S$ be the swarm's barycenters, the objective function of the $\ell$-th swarm is given by

$$\mathcal{E}^\ell(\nu) := \mathrm{KL}\left(\frac{1}{n_S}\nu + \frac{1}{n_S - 1}\sum_{h \neq \ell}\overline{\mu}^{\alpha,h} \,\middle|\, \mu^{\mathrm{targ}}\right) \, .$$

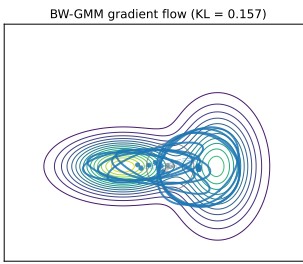 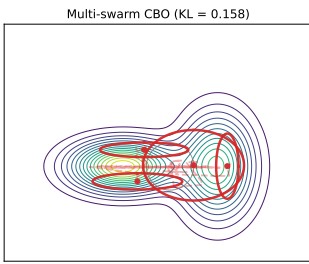 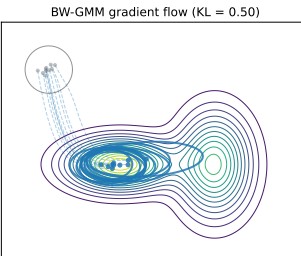 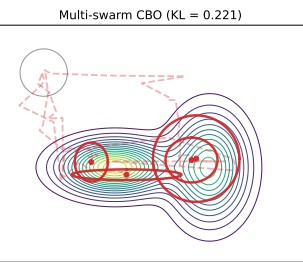

*Figure 9.* **Extension to GMM via multi-swarm approach.** Validation of the multi-swarm CBO approach and comparison with the GMM approximation of the Wasserstein Gradient Flow proposed in (Lambert et al., 2022). In the multi-swarm strategy, the particles are divided into $n_S = 4$ swarms, each with $N = 10$ Gaussian particles; each swarm evolves by an internal Gaussian CBO dynamics while its objective depends on the barycenters of the other swarms, thereby encouraging the swarms to cover different regions of the target. For the GF, we use $N = 10$ particles. The left plots correspond to an initialization close to the modes of the target measure, while the right plots correspond to an initialization far from the modes. Parameters used are the same as $d = 2$ single-swarm experiments. The tests show that CBO-type dynamics is able to find better or comparable approximations of the target measure in terms of KL divergence (see final KL values in the plots' titles).

Intuitively, the $\ell$-th swarm should find the best Gaussian measure $\nu$ that, when summed up with the other swarm's barycenters $\overline{\mu}^{\alpha,h}$, $h \neq \ell$, provides the best match with respect to the given target $\mu^{\text{targ}}$. Note that, to define the barycenters one needs the objective function $\mathcal{E}^\ell$, which, in turns, requires knowledge of the barycenters, so the definition appears to be ill-posed. In practice, the algorithm starts from unweighted barycenters at step $k = 1$ to define the swarms' objective functions at $k = 1$, and then the objects can be defined recursively. The precise algorithmic strategy is described in Algorithm 2 and a validation test is presented in Figure 9.

---

**Algorithm 2** Multi-swarm Gaussian Consensus-Based Optimization

---

**Input:** Target $\mu^{\text{targ}}$, reference $\mu^0 = \mathcal{N}(0, \Sigma^0)$
        parameters $\lambda = 1, \sigma, \Delta t > 0, \alpha \gg 1$, number of swarms $n_S \in \mathbb{N}$, swarm size $N \in \mathbb{N}$
Initialize particles $(m^{\ell,i}, T^{\ell,i}) \in \mathbb{R}^d \times \mathbf{Sym}_d$, $i \in [N]$, $\ell \in [n_S]$
For each swarm $\ell$, compute the initial unweighted barycenter $\overline{\mu}^{0,\ell}$
**repeat**
    **for (parallel)** $\ell = 1$ **to** $n_S$ **do**
        Define the swarm-dependent objective (E)
        Evaluate $\mathcal{E}^\ell(\mu^{\ell,i})$ with $\mu^{\ell,i} = \mathcal{N}(m^{\ell,i}, \exp_{\Sigma^0}(T^{\ell,i}))$
        Set swarm weights $\omega^{\ell,i} \propto \exp(-\alpha \mathcal{E}^\ell(\mu^{\ell,i}))$
        Compute swarm consensus $(\overline{m}^{\alpha,\ell}, \overline{T}^{\alpha,\ell})$ with $\{\omega^{\ell,i}\}_{i=1}^N$
        Set $\overline{\mu}^{\alpha,\ell} = \mathcal{N}(\overline{m}^{\alpha,\ell}, \exp_{\Sigma^0}(\overline{T}^{\alpha,\ell}))$
    **end for (parallel)**
    **for (parallel)** $\ell = 1$ **to** $n_S$ **do**
        **for (parallel)** $i = 1$ **to** $N$ **do**
            Update particle $(m^{\ell,i}, T^{\ell,i})$ as in single-swarm algorithm using swarm consensus $(\overline{m}^{\alpha,\ell}, \overline{T}^{\alpha,\ell})$
        **end for (parallel)**
    **end for (parallel)**
**until** convergence reached
**Output:** Mixture approximation $\overline{\mu}^\alpha = (1/n_S) \sum_{\ell=1}^{n_S} \overline{\mu}^{\alpha,\ell}$

---

