# OpenReview forum: "Variational inference via Gaussian interacting particles in the Bures-Wasserstein geometry"
_ICML.cc/2026/Conference — ICML 2026 regular_

### Official Review · Reviewer_3oZk · 2026-03-06

**Soundness:** 3
**Presentation:** 3
**Significance:** 3
**Originality:** 3
**Overall Recommendation:** 5
**Confidence:** 4

**Summary:**

- This paper addresses Gaussian variational inference (GVI), where the variational distribution is restricted to the family of Gaussian distributions.
- The main idea is based on consensus-based optimization (CBO), where  a set of particles is maintained and each particle represents a Gaussian. Each particle is stochastically moved towards a consensus point, defined as a weighted average of the particles, which serves as a proxy for the global minimizer.
- the authors also propose the linearized Bures-wasserstein metric (LBW) as a computationally efficient way to compare Gaussian particles and compute the weighed average used in CBO.
- experiments on several toy datasets are shown to illustrate the superiority of the proposed method.

**Compliance With Llm Reviewing Policy:**

Affirmed.

**Final Justification:**

Authors have addressed my concerns.

**Key Questions For Authors:**

See the weaknesses.

**Limitations:**

The experiments could be improved further by comparing with other existing Gaussian VI methods.

**Strengths And Weaknesses:**

Strengths:
- This paper presents an interesting idea: using consensus-based optimization (CBO) to approach the global minimizer, while existing gradient-based Gaussian VI methods typically rely on the convexity assumption on the objective function.
- Applying CBO to Gaussian VI requires the expensive computation of the barycenter of Gaussians (i.e. the weighted average in Bures-Wasserstein space). The authors propose using the LBW metric to enable a more efficient computation, which is reasonable practical approximation.
- the paper is well written and structured. It is easy to follow.


Weaknesses:
- In subsection 2.1, the authors mentioned that adding noise to Gaussian particles on the BW manifold may lead to ill-posed dynamics. However, it is unclear that why this issue can be resolved by working in the linearization of BW space. A more detailed explanation would be helpful.
- It is not clear whether the proposed method can be extended to gaussian mixtures, which are commonly used to approximate complex targets. Some existing works already address this setting (see below), and a discussion of whether the proposed approach can handle such cases would be helpful.
- Gaussian VI has been studied in some work (see below). The experimental comparison with only the method of Lambert et al., seems incomplete. Including comparisons or discussions with additional approaches in single Gaussian and mixture of Gaussian would provide a more comprehensive evaluation.

[1] W. Lin et al., Fast and Simple Natural-Gradient Variational Inference with Mixture of Exponential-family Approximations, ICML 2019.\
[2] Nguyen et al., Wasserstein gradient flow over Variational parameter space for variational inference, Aistats 2025.\
[3] Lambert et al., Variational inference via wasserstein gradient flows, NeurIPS 2022.

---

> ### Author Rebuttal · Authors · 2026-03-30
>
> We thank the reviewer for the comments.
>
> 1. The problem with the noise in the BW space is that the Riemannian manifold is not complete. This means that if we move along a random direction, we might land on the boundary of the BW space, at which the Riemannian geometry is not well defined. This would prevent us from sampling another random tangent direction altogether and iterating the random walk. On the contrary, when we linearize the space, the geometry is fully determined by the reference measure used for the linearization. Specifically, the space reduces to a Hilbert space with product defined by the Riemannian metric at the linearization point. Most importantly, the space is now geodesically complete. We plan to stress this important aspect more in the manuscript.
>
> 2. We have designed a proof of concept that extends the current approach to obtain Gaussian mixture approximations. This is achieved via a multi-swarm approach where the swarms interact with each other through their barycenters. We refer to the reply to Reviewer pRzP for more details on the multi-swarm approach. A validation experiment is included in https://figshare.com/s/5cc9f909bab8d7331a8d , Figure 3.
>
> 3. We have performed more experiments by considering more baseline algorithms, in particular Gaussian SVGD and natural (Fisher--Rao) gradient descent. See Figures 1 and 2 in https://figshare.com/s/5cc9f909bab8d7331a8d . We have also included the GMM variants of the Bures--Wasserstein gradient flows for a richer comparison. Results show that the proposed CBO method is competitive, especially in the multi-modal target distribution scenarios.

---

> > ### Author Rebuttal · Reviewer_3oZk · 2026-04-02
> >
> > Thanks for the response.
> > Authors have addressed my concerns. I have updated my score accordingly.

---

### Official Review · Reviewer_5xat · 2026-03-09

**Soundness:** 4
**Presentation:** 4
**Significance:** 3
**Originality:** 3
**Overall Recommendation:** 5
**Confidence:** 4

**Summary:**

The paper introduces an extension of the consensus dynamics of (Borghi et al. 2025) to a stochastic dynamics on the Bures-Wasserstein manifold of Gaussian measures. The dynamics is supposed to determine the global mode of a target distribution via Gaussian particles that explore the landscape of the log-exponential transformed KL-objective function and eventually reach consensus. The stochastic process is shown to be well-posed and a convergence result from (Carillo et al. 2021) is extended accordingly.

**Compliance With Llm Reviewing Policy:**

Affirmed.

**Key Questions For Authors:**

(1) How is the reference Gaussian 0-measure chosen in practice? For example, if you wish to compute the barycenter of n Gaussian distributions, how do you choose the reference measure to obtain a unique barycenter? The weak dependence on the reference measures illustrated by Figure 6 and the finding “no systematic improvement” can be achieved when the linearization is updated (Section 4.2, lines 382/3) is puzzling. How can this be explained?

(2) Please respond to the above points Weaknesses: (1) and (2).

**Limitations:**

See Weaknesses, point (2) above.

**Strengths And Weaknesses:**

Strengths
The paper is mathematically rigorous and explains very well the emergence of the geometric set-up in the research area in optimal transport. The approach is broadly applicable and, by exploiting the linearized Wasserstein geometry, tames to some extent the computational costs of geometric averaging. Experimental results for toy problems demonstrate the beneficial effect of the stochastic part of the dynamics from the viewpoint of nonconvex optimization.


Weaknesses

(1) A discussion of the assumptions ot Theorem 3.5 is missing. Are there  practically relevant cases in which the assumptions can be checked? The statement “in CBO algorithms one typically fixes … parameters are set to …” (Section 4.2, 2nd paragraph) seems to ignore Theorem 3.5. How were the parameters set in the experiments? Do these settings reflect Remark 3.6?

(2) The paper does not discuss limitations of the approach. The performance of the consensus point \tilde(z) as characterized by Theorem 3.5 looks too good since the terms of the empirical objective bound vanish with increasing alpha, and alpha is chosen quite large in the experiments. Where and how does the "no free lunch theorem" show up? Providing a failure case which demonstrates would make the paper (even) stronger. The outlook section says “… would allow to handle broader classes of probability measures …”: which measures cannot be handled with the proposed approach?

(3) Although the presentation of the ingredients of the approach and closely related prior works is fine, I am missing a somewhat broader perspective on related work. Papers with the same scope and utilizing Gaussian measures, yet of course not the proposed particle process, should be taken into account. See, e.g., Ollivier et al, “Information-Geometric Optimization Algorithms: A Unifying Picture via Invariance Principles”, JMLR Vol 18, 2017.

---

> ### Author Rebuttal · Authors · 2026-03-30
>
> We thank the reviewer for the remarks.
>
> 1. In our experiments, the Gaussian reference measure is chosen to be the standard one, $\mathcal{N}(0,I_d)$. In the algorithm, it is fixed so that it is always possible to define a unique linearized barycenter for all Gaussians. We conjectured that updating the reference measure  during the computation might improve the results, as it provides a more accurate linearization of the Riemannian manifold. In this case, the new reference measure was the barycenter computed at the previous time step. Updating the reference measure, though, did not bring any benefit in the experiments considered. We conjecture that this might be due to the fact that, in these toy experiments, linearizing at $\mu^0 = \mathcal{N}(0,I_d)$ does not introduce any major distortion, and that the repeated computation of Riemannian logarithmic and exponential maps (needed for the reference measure update) might actually numerically deteriorate the computed solutions.
>
>
> 2. Regarding ``Weaknesses, point (1)'': We agree with the reviewer that additional discussion is needed regarding the theorem's assumptions in Remark 3.6. The two key aspects of the assumptions are that $2\lambda > \sigma^2$ and that $\textup{Var}(\rho_0)$ should be small. The first one comes from an intrinsic property of the particle dynamics, and says that, for consensus emergence to appear, the diffusion parameter $\sigma$ should not be too large. For $\lambda = 1$, the direct application of the rule would require $\sigma<\sqrt{2}$. Empirically, we found this condition to be too restrictive, as the algorithm leads to good results up to $\sigma \approx 6$, in the problems considered.
> The need for $\textup{Var}(\rho_0)$ to be small, instead, lies more in the variance-based proof strategy rather than in the particle dynamics. Indeed, such a restriction is not present when using the different analysis strategy proposed in [Fornasier, Klock, Riedl 2024].
>
> 3. Regarding ``Weaknesses, point (2)'': The no-free-lunch theorem shows up in the fact that the analysis is performed for the mean-field approximation, while the algorithm uses a finite number of particles $N$, and the quantitative error of such an approximation deteriorates for $\alpha \gg 1$ (see [Gerber, Hoffmann, Kim, Vaes 2025]). Therefore, Theorem 3.5 requires large $\alpha$, which in turn requires large $N$ to have a good mean-field approximation, leading to high computational cost. We plan to include this discussion in Remark 3.6.
> The current method is limited to the space of Gaussian approximations, and therefore cannot approximate well target measures that are very different from Gaussian. In additional experiments, we extended  the dynamics to Gaussian Mixture Models in the sense that the output of the algorithm is a GMM (as done in [Lambert at al. 2022]), not that the target measure is a GMM.
> This is done via a multi-swarm approach, see answer to Reviewer pRzP for more details.
> In any case, the target measure can, also in the single-Gaussian case, be any measure.
> More generally, regarding the limitations, we consider the biggest one to be the tuning of the noise in higher-dimensional problems. We plan to underline this limitation more clearly in the manuscript.
>
> 4. Regarding ``Weaknesses, point (3)'': As suggested by the reviewer, and by the others, we plan to include more baseline algorithms to allow for a better comparison with other methods, see https://figshare.com/s/5cc9f909bab8d7331a8d . We decided to include algorithms that are also based on the evolution of Gaussian measures, but with different geometries. In particular, we have included Gaussian SVGD and natural (Fisher--Rao) gradient descent.

---

> > ### Author Rebuttal · Reviewer_5xat · 2026-04-02
> >
> > Thanks for the response. Authors address the relevant points well. I keep my rating.

---

### Official Review · Reviewer_pRzp · 2026-03-10

**Soundness:** 3
**Presentation:** 2
**Significance:** 2
**Originality:** 2
**Overall Recommendation:** 4
**Confidence:** 3

**Summary:**

The authors study variational infernece by optimizing over the space of Gaussian probability measures. It proposes a new zero-order optimization method based on interacting Gaussian particles that evolve according to a CBO mechanism in a linearized Bures-Wasserstein geometry. This paper establish theoretical properties such as well-posedness and onvergence via mean-field analysis and experiments on Gaussian variational inference tasks show that the method can outperform gradient-based approaches.

**Compliance With Llm Reviewing Policy:**

Affirmed.

**Final Justification:**

After rebuttal, my concerns are addressed, and I increase score to 4.

**Key Questions For Authors:**

See the three questions in the weakness part.

**Limitations:**

The authors did not discuss the limitations.

**Strengths And Weaknesses:**

Strengths:

The paper is mathematically solid in the theory of optimal transport and variational inference. The proposed algorithm is derived from the Bures–Wasserstein geometry of Gaussian measures, and the theoretical development appears rigorous (although I did not check it line by line).

Weaknesses:

(1) The experimental evaluation is relatively limited. Most experiments are conducted on low-dimensional synthetic Gaussian mixture models, which makes it difficult to assess the practical usefulness of the proposed method in realistic variational inference settings.

(2) The set of baselines is rather small. The experiments mainly compare against a Wasserstein gradient flow method, while many other commonly used variational inference or particle-based inference methods are not included. For example, full-covariance Gaussian VI, SVGD, Stein Variational Newton.

(3) The scalability of the proposed approach is not thoroughly studied. Although experiments in moderate dimension are included, it remains unclear how the algorithm behaves in higher-dimensional inference problems. Can the authors provide a separate experiment section to analyize the scalability?

---

> ### Author Rebuttal · Authors · 2026-03-30
>
> We thank the reviewer for the comments.
>
> We agree that the experiments are limited to the approximation of synthetic low-dimensional targets and lack applications to more realistic settings. To address point (2), we have performed comparisons with more baseline models, including Gaussian SVGD and natural (Fisher--Rao) gradient descent. The results show that the other baseline methods, being essentially first-order methods, may also get stuck in sub-optimal solutions due, we conjecture, to the lack of suitable explorative behavior. In $d = 10$, SVGD returns on average better results, but requires a much smaller time-step $\Delta t /20$ for stability. This is illustrated in Figures 1 and 2 in https://figshare.com/s/5cc9f909bab8d7331a8d
>
> As discussed in Remark D.1, the main issue with higher-dimensional experiments is the tuning of the noise parameter $\sigma$, as large values might lead to divergence, while small values may lead to premature convergence. Also, for large $d$, the role of the covariance overwhelms that of the mean, as the number of covariance parameters scales as $d^2$. A fruitful approach might be to restrict the Bures--Wasserstein geometry to the space of isotropic Gaussians (or to diagonal covariance matrices), as done in [Petit-Talamon, Lambert, Korba 2025]. We plan to remark further on this current limitation in the main text.
>
> Still, we believe that our contribution, even if limited to synthetic test problems, underlines a limitation of first-order methods in the presence of non-convex problems and opens the possibility of novel stochastic particle dynamics to address them. Also, the paradigm can be extended to Gaussian Mixture Models via a multi-swarm approach showing the flexibility of the approach.
>
> Description of multi-swarm approach. The particle system is partitioned into $n_S$ sub-swarms, each evolving according to a Gaussian CBO dynamics analogous to the single-swarm case, but with a swarm-dependent objective. Denoting by $\overline{\mu}^{\alpha,\ell}$, $\ell=1,\dots,n_S$, the swarm barycenters, the objective for the $\ell$-th swarm is
> $
> \mathcal{E}^\ell(\nu) := \mathrm{KL} ( (1/{n_S})\nu + 1/(n_S-1)\sum_{ h \neq \ell} \overline{\mu}^{\alpha, h}  |  \mu^\mathrm{targ} ).
> $
>
> Thus, the $\ell$-th swarm seeks a Gaussian measure $\nu$ whose combination with the barycenters of the other swarms best matches the target $\mu^{\mathrm{targ}}$. Although this definition is recursive, since $\mathcal{E}^\ell$ depends on the barycenters, which themselves are defined through $\mathcal{E}^\ell$, the scheme is initialized using unweighted barycenters at step $k=1$ and then iterated recursively. Figure 3 in additional experiments (https://figshare.com/s/5cc9f909bab8d7331a8d) validates the approach, which is inspired by multi-objective particle optimization [Klamroth, Stiglmayr, Totzeck (2024)].

---

> > ### Author Rebuttal · Reviewer_pRzp · 2026-04-04
> >
> > Thanks for the rebuttal. My concerns are addressed. I increase the score to 4.

---

### Official Review · Reviewer_wic8 · 2026-03-11

**Soundness:** 2
**Presentation:** 3
**Significance:** 2
**Originality:** 3
**Overall Recommendation:** 4
**Confidence:** 2

**Summary:**

This paper focuses on the optimisation problem in Gaussian variational inference. Specifically, this paper optimizes in Gaussian family for a distribution that minimizes a target loss function, of which one typical case considered is the reverse KL. The authors claims that the current one-order optimization under Bures–Wasserstein (BW) geometry can easily get stuck in locla optimum and
proposes a Consensus-Based Optimization (CBO) based method to solve this problem. The authors build firstly a Linearized Bures–Wasserstein (LBW) representation and parametrize the Gaussian family into a linear space. Then, the authors define a system of multi gaussian particles, showing its well-posdness and proving their convergence to gloabal minimum.
In experiments on several synthetic variational inference problems of synthetic Gaussian mixture setting, compared with BW gradient-flow baseline, the Gaussian CBO converges faster and demonstrate a smaller average KL-divergence.

**Compliance With Llm Reviewing Policy:**

Affirmed.

**Final Justification:**

The rebuttal is helpful and clarifies some points. I also appreciate that the code will be released. However, my main concerns remain. First, the practical relevance of the method is still limited by the restriction to the single-Gaussian family. Second, the paper still does not compare with a standard stochastic first-order optimizer, which I think is an important baseline for judging the practical value of the proposed approach, but I understand the time constraints during the rebuttal process. I keep my score unchanged.

**Key Questions For Authors:**

1. For practical relevance, could the authors clarify in which practically relevant scenarios optimizing within a single Gaussian family is itself an important objective? especially compared with approaches based on  richer families, such as mixtures of Gaussians? e.g. [Petit-Talamon et al. 2025]?

2. Could the authors clarify how the regularization $\epsilon$ affects the interpretation of the convergence result for the original KL objective?

3. A natural alternative of minimizing a target loss (such as KL) is through reparameterization and stochastic-gradient optimization. For example, you could write $X=m+L\xi, \xi\sim\mathcal N(0,I), \Sigma=LL^\top$ and estimate (just as in your paper, you have estimated the $V$ by a quadrature rule) a gradient for the same target $KL\!\left(\mathcal N(m,LL^\top)\,\middle\|\,\mu_{\text{targ}}\right)$, and then apply common stochastic first-order optimizer such as SGD and Adam to optimize. Since the proposed method relies on repeated objective evaluations, have the authors considered including such a baseline or explain why gradient-free is preferable?

4. Is it possible to discuss the running time comparison or time complexity analysis?

5. For the baseline method GF, are there any hyperparameters that need to be tuned? Since the paper analyzes the sensitivity of the proposed method with respect to $\sigma$, $N$, and the choice of reference, I wonder whether the GF baseline also involves hyperparameters that should be tuned to ensure a fair comparison.

6. A question about Figure 5, why the confidence interval (blue shaded area) of GF remains constant?

**Limitations:**

Yes.

**Strengths And Weaknesses:**

Strengths:

1. The problem setting and motivation are clear. And the paper is easy to follow and understand. The LBW representation is very interesting and is the main contribution of this paper.

2. The theoretical part is clear and complete. From the well-posedness to the global convergence, the proof shows that this method CBO is not a purely heuristic method.

3. Experiments (partially) supports the main point for the convergence.

Weaknesses:

1. The whole method is limited in single Gaussian family. The optimization space is a single gaussian, thus limiting its generality to multimodal or skewed distribution. The contribution of this paper is mainly for an optimizer in a restricted family, thus limiting its practical impact.

2. The convergence argument appears to rely heavily on existing Euclidean mean-field CBO theory after moving to the LBW parameterization.  For the bridge from BW to LBW,  for KL-type objectives, the regularity assumptions of Theorem 3.5 seem to require the clipped/regularized objective $KL_\epsilon$, rather than the original KL directly. The relation between the minimizers of the regularized objective and those of the original KL remains a gap.

3. Experiments is not sufficient and only limited on synthetic Gaussian mixture targets. It would be better to include: more complexe Bayesian posterior tasks, more VI baselines models as well as algorithm complexity analysis.

4. Even if the multi gaussian family is not in the scope of this paper,  it would still be useful to at least discuss, and ideally compare with some multi-gaussian family method, such as  Variational Inference with Mixtures of Isotropic Gaussians (Petit-Talamon, Lambert, and Korba, NeurIPS 2025). This would help to understand the advantage and impact of the proposed method.

---

> ### Author Rebuttal · Authors · 2026-03-30
>
> We thank the reviewer for the report.
>
> 1.  The accuracy of single-Gaussian approximations for VI problems has been studied in [Katsevich, Rigollet 2024] and [Spokoiny, Panov 2025], showing their suitability for high-dimensional problems. In lower-dimensional settings, we agree with the reviewers that Gaussian mixtures (GMMs) provide a more expressive class over which to solve VI problems. As in other papers considering similar methods, we started with the single-Gaussian case because of its rich geometry, which allowed us to define a stochastic particle dynamics. In the currently proposed CBO algorithm, particles reach full consensus around a single point, leading to a single-Gaussian output. This novel framework suggests that Gaussian VI problems can be solved using dynamics that differ from the usual deterministic gradient-descent-type methods. We are currently extending the method to return a GMM by using a multi-swarm approach (see response to Reviewer pRzp)
>
> 2. The $\varepsilon$-regularization of the KL divergence only affects the function value for Gaussians that are close to being singular. Therefore, the regularization affects the original problem only if the solution to the non-regularized problem is a Gaussian with an almost degenerate covariance. We plan to emphasize this point more clearly in the paper.
>
> 3. The parametrization $(m,L)$ for the Gaussian $\mathcal{N}(m, LL^\top)$ endowed with the Euclidean geometry is indeed a natural choice. At the covariance level, however, this parametrization induces exactly the same evolution as the BW flow, because the BW geometry can be viewed as arising from the Euclidean geometry on matrix square roots through the map $L \mapsto LL^\top$; see [Malag\'o et al. 2018] and [Bhatia et al. 2019]. This is also consistent with the square-root formulation used in [Lambert et al. 2022] for the BW gradient flow. Regarding the use of a stochastic first-order optimizer, we assume the reviewer means ``stochastic'' in the sense that the KL divergence is computed using Monte Carlo samples from the Gaussian iterate, rather than quadrature rules. We agree that noisier objective-function evaluations may lead to more exploratory behavior in first-order methods, but this would be better viewed as an implementation strategy rather than as genuine noise injection at the level of the dynamics. We also compared the algorithm against additional baseline methods (Gaussian SVGD and natural/Fisher--Rao gradient descent). We kept the KL evaluation based on quadrature rules in order to better compare the different optimization dynamics. See Figures 1 and 2 in https://figshare.com/s/5cc9f909bab8d7331a8d
>
>
> 4. The baseline algorithms only require the choice of a step size $\Delta t$. If the potential $V$ of the target measure $\mu^{\mathrm{targ}} \propto e^{-V}$ has steep gradients, then the step size may need to be small, or a different time-integration strategy may be used. In CBO, $\Delta t$ does not play a comparably relevant role in the dynamics, so in the experiments we chose $\Delta t$ small enough to ensure that the baseline algorithms remain stable. Therefore, as pointed out by the reviewer, CBO does require tuning more parameters. We would argue, however, that the only key parameter is the noise strength $\sigma$, since $\alpha$ is always chosen to be very large, the number of particles depends mainly on the computational budget, and the choice of reference measure can in principle be adapted to the current particle population. By contrast, tuning $\sigma$ is more challenging and may depend on the dimension. This is analogous to tuning the annealing schedule in simulated annealing and, more generally, may be viewed as the price to pay for an algorithm with additional exploratory stochasticity. We plan to underline this aspect more clearly in the final version.
>
> 5. We conjecture that the confidence interval remains constant because the algorithm runs have already reached convergence. The median appears to become constant at a slightly later time, but this minor improvement is not reflected in the confidence interval, since the runs that ended in a suboptimal solution exhibit only very limited further improvement.

---

> > ### Author Rebuttal · Reviewer_wic8 · 2026-04-02
> >
> > Thank you for the reply. I think several key issues are still unresolved. First, the practical relevance of the work remains limited by the restriction to the single-Gaussian family. Second, I believe that a standard reparameterized first-order optimizer with stochastic noise (e.g. Monte Carlo noise, as mentioned by the authors) should be included as an important baseline. The practical question is whether a stochastic first-order optimizer based on Monte Carlo gradient estimation, such as Adam, could already achieve similar performance. Will the code be released? Overall, I think some core concerns remain unresolved, and I believe my original score is fair.

---

> > > ### Author Response · Authors · 2026-04-05
> > >
> > > Thank you the remarks. Due to time constraints of the review process we have been unfortunately able to add further baseline comparisons beyond the ones added during the first round. The point raised by the reviewer is indeed interesting and will be mentioned as a natural follow up comparison. It is in line with other literature where stochastic evaluations of gradients are used to favor exploration, and not simply as a way to reduce computational cost (see e.g. [Tankala-Nagaraj-Raj COLT 2025]).
> > >
> > > The code will be released, and now is also already available at the link previously provided.

---

### Decision · Program_Chairs · 2026-04-30

**Decision:**

Accept (regular)

**Comment:**

This paper proposes a consensus-based optimization (CBO) approach for Gaussian variational inference under a linearized Bures–Wasserstein (LBW) parameterization. The work is mathematically well-developed, with rigorous analysis of well-posedness and convergence, and reviewers consistently recognize the technical depth, clear geometric formulation, and the novelty of adapting CBO dynamics to Gaussian measures.

At the same time, the practical scope of the method is limited. The approach is restricted to the single Gaussian family, which constrains its applicability in realistic variational inference problems involving multimodal or complex posteriors. While extensions to Gaussian mixtures are discussed in the rebuttal, they are not part of the main contribution and remain preliminary.

The empirical evaluation is also limited. Experiments are largely conducted on low-dimensional synthetic tasks, and although additional baselines were included during rebuttal, the evaluation does not cover realistic Bayesian inference problems. Moreover, the lack of comparison with standard stochastic gradient-based VI methods makes it difficult to assess the practical advantages of the proposed approach.

Finally, concerns remain regarding scalability and robustness, including sensitivity to hyperparameters (notably the noise parameter) and behavior in higher-dimensional settings. While some of these points are acknowledged in the rebuttal, they are not fully resolved. Overall, despite strong theoretical contributions, these limitations reduce the paper’s practical impact.